# Actively Testing Your Model While It Learns: Realizing Label-Efficient Learning in Practice

**Dayou Yu**[1]    **Weishi Shi**[2]    **Qi Yu**[1*]

Rochester Institute of Technology, Rochester, NY 14623[1]
University of North Texas, Denton, TX 76203[2]
{dy2507,qi.yu}@rit.edu[1]    weishi.shi@unt.edu[2]

## Abstract

In active learning (AL), we focus on reducing the data annotation cost from the model training perspective. However, "testing", which often refers to the model evaluation process of using empirical risk to estimate the intractable true generalization risk, also requires data annotations. The annotation cost for "testing" (model evaluation) is under-explored. Even in works that study active model evaluation or active testing (AT), the learning and testing ends are disconnected. In this paper, we propose a novel active testing while learning (ATL) framework that integrates active learning with active testing. ATL provides an unbiased sample-efficient estimation of the model risk during active learning. It leverages test samples annotated from different periods of a dynamic active learning process to achieve fair model evaluations based on a theoretically guaranteed optimal integration of different test samples. Periodic testing also enables effective early-stopping to further save the total annotation cost. ATL further integrates an "active feedback" mechanism, which is inspired by human learning, where the teacher (active tester) provides immediate guidance given by the prior performance of the student (active learner). Our theoretical result reveals that active feedback maintains the label complexity of the integrated learning-testing objective, while improving the model's generalization capability. We study the realistic setting where we maximize the performance gain from choosing "testing" samples for feedback without sacrificing the risk estimation accuracy. An agnostic-style analysis and empirical evaluations on real-world datasets demonstrate that the ATL framework can effectively improve the annotation efficiency of both active learning and evaluation tasks.

## 1  Introduction

Labeled data are essential for supervised learning in both model training and evaluation. Active learning (AL) provides a promising direction to reduce the human annotation cost by constructing a smaller but more effective labeled dataset for training purposes [24, 27, 12, 5, 1, 8]. However, AL only partially addresses the human annotation cost as the cost of constructing a labeled dataset for model testing has been overlooked. In reality, the annotation budget could easily be drained by building large test datasets that most modern AL methods rely on to evaluate the model performance and determine the stopping criterion for learning. In contrast to AL, active model evaluation or active testing (AT) focuses on actively selecting the testing data. Few existing efforts develop unbiased risk estimation techniques for label-efficient model evaluation [20, 14]. However, these methods are designed to evaluate fixed models that have been fully trained, making them ill-suited for evaluating an actively learned model, which is constantly updated and inadequately trained during most parts of the learning process. To address these fundamental challenges and realize label-efficient learning in practice, we propose to integrate AT with AL in a novel and efficient learning framework, referred to

---

*Corresponding author.

37th Conference on Neural Information Processing Systems (NeurIPS 2023).

as ATL. The ATL framework systematically addresses the unique active testing challenges arising from evaluating an actively learned model that is under-trained and constantly evolving. It leverages an interactive learning-testing-feedback process to better control the overall labeling budget, which not only achieves efficient evaluation of the AL model but also ensures faster convergence in model training. Through periodical testing, ATL gains useful insights to terminate the learning process as early as possible that avoids allocating unnecessary labeling budget.

Incorporating AT into AL is a highly non-trivial task. Unlike AL, AT has two inter-connected objectives: i) design of an unbiased risk estimator to quantify the model performance, and ii) design of a sampling strategy to select informative test samples. Achieving both objectives simultaneously under the AL setting with a constantly evolving and under-trained model is far more challenging than existing active model evaluations that assume a fixed and adequately trained

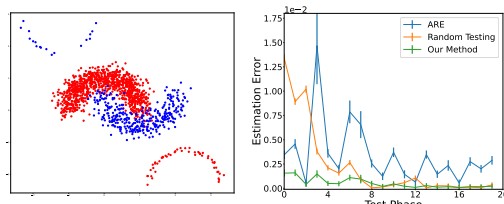

(a) Data distribution    (b) Active estimation error

Figure 1: Risk estimation comparison

model. For the first objective, its challenge stems from the potential biased active test samples brought by the second objective. We need an unbiased risk estimator to ensure the fairness of the evaluation. An unbiased risk estimator can be achieved by importance sampling or its variants [25, 20, 21, 22, 19, 16]. Recent active testing works also employ expectation analysis and propose other weighting mechanisms [10, 14]. However, all these works assume that the model being evaluated is fixed and already well-trained [20, 14]. Therefore, these estimators are not designed to support a dynamic AL setting, where the model continues to evolve when learning from newly labeled samples. Figure 1 shows that both the standard estimator (*i.e.,* random testing) and direct adaptation of an existing unbiased estimator (ARE) [20] fail to provide an accurate and consistent model evaluation in the more challenging AL setting. As for the second objective, different from sample selections in AL, the testing selection needs to be compatible with the unbiased risk estimator. It is essential to ensure asymptotic convergence that guarantees an unbiased model evaluation while reducing the variance that can improve the convergence speed and provide a stable signal to terminate the learning process as early as possible.

Conducting AT along with AL enables ATL to seamlessly connect the active training and testing sides, which allows them to communicate with and guide each other within a well-integrated learning process. To this end, the proposed ATL framework leverages an active learning-testing-feedback (LTF) loop, which largely resembles real-world human learning. In particular, within the LTF loop, the active testing results are kept after each active learning round, which mimics the quizzes in the human learning setting. We refer to them as *active quizzes*. As shown in Figure 2, the

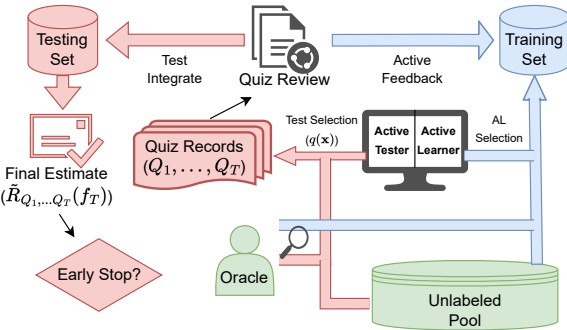

Figure 2: Overview of the ATL Framework

active learner acts as a student while the active tester acts as a teacher. Both the student and the teacher select data from the unlabeled pool and get their labels from the oracle. The student attempts to use the labeled data to pass the quizzes provided by the teacher. On the other hand, the teacher provides a fair quiz to evaluate the student's performance. Meanwhile, the teacher also sends back the keys of some quiz questions, which are referred to as *active feedback*, to help the student learn better.

Given a dynamic AL process, multiple active quizzes will be created at different learning phases. ATL forms an integrated risk estimator by aggregating all the quizzes. Our theoretical analysis guarantees that the integrated risk estimation converges to the true risk asymptotically. The aggregation strategy also minimizes the variance, which ensures faster convergence. Furthermore, through active feedback, the AL model can benefit from training with a small proportion of the testing samples (as feedback) after each quiz while maintaining the asymptotic convergence condition of the risk estimation. Finally, the integrated risk estimator provides an unbiased estimate of the model performance with a small variance, which provides a convenient way to determine whether the AL model has converged.

This can further save the labeling budget by stopping the learning early. We may also combine the estimated risk with other available information from the unlabeled data to create more systematic early-stopping criteria. Our main contributions are summarized as follows:

- the first ATL framework that builds an active learning-testing-feedback loop to mimic human learning for achieving label-efficient and dynamic evaluation of actively learned models,
- a theoretically sound integrated risk estimator that combines active quiz results and minimizes the variance of the difference between the estimated and true risks,
- an innovative active feedback process to further improve learning effectiveness without extra labels,
- a systematic early stopping criterion that combines integrated risk estimation with information of unlabeled data to terminate AL process early.

We conduct experiments on both synthetic and real-world datasets to show the improved testing performance of the proposed ATL framework.

## 2 Related Work

**Active model evaluation.** Actively testing the model performance in a label efficient way is relatively under-explored. Among the few excising efforts, model testing is treated independently from model training, which is unrealistic in most practical settings. The problem of active risk estimation (ARE) was first formulated in [20], which used importance sampling to construct an unbiased risk estimator. Then, the optimal test sampling distribution is derived from an optimization problem that minimizes the variance of the risk error. A similar process has been used to estimate more complex loss measures [21] or perform active model selection [23]. However, such risk estimator is no longer unbiased nor label-efficient when an AL process is involved that samples from the same unlabeled pool for model training. Farquhar et al. propose two risk estimators to cancel the statistical bias by active data sampling strategy in a data-efficient manner [10]. However, their proposed estimators could not deal with the overfitting bias making them less suitable for model evaluation.

In a more recent work, Kossen et al. propose to use the unbiased risk estimators to perform active testing [14]. The difference from [20] is that the selection order of the testing points is considered, and the weights are defined for selection indices instead of a single testing proposal distribution. In this way, the testing proposal can be evaluated at each round, making the setting more realistic. However, the optimal proposal is still impossible to obtain. To approximate the optimal proposal, a surrogate model is needed, which is similar to the introspective risk model in [20]. The surrogate model idea is further extended in [15], where the unbiased risk estimator is replaced by the predicted loss from the surrogate model. Yilmaz et al. propose to replace the commonly used importance sampling with Poison sampling, which better stabilizes the test samples [29]. A different line of work leverages stratified sampling to first stratify the unlabeled pool and then conduct efficient sampling over strata in an importance sampling manner [13, 3, 30]. However, all those work is limited to offline test sampling in which the test proposal is fixed during the sampling process and could not benefit from prior knowledge such as the previously labeled data.

**Early stopping in AL.** Early stopping strategies in AL have been sparsely investigated over years [2, 7, 6, 31, 26, 18]. Most methods only consider the learning process and do not consider a fully connected learning-testing loop. Some have shown that the stopping criteria based on unlabeled information are more effective than those based on labeled information. This inspires us to combine the proposed integrated risk estimation with unlabeled data information to form a systemic early stopping criterion to further reduce the labeling budget.

## 3 The ATL Framework

**Problem setting.** We formulate the ATL framework under a novel and fully integrated learning and testing-while-it-learns setting. W.l.o.g., we assume that all data are generated from an underlying distribution, and we could obtain the label $y$ through the true labeling function: $y = h(\mathbf{x})$ provided by the human annotator (note that unlike the predictive model $f_\theta(\cdot)$, the true labeling function is not governed by the parameter $\theta$ and it might be non-realizable). Similar to a traditional pool-based AL setting, we have a small labeled set $\mathcal{S}_L = \{(\mathbf{x}, y)\}^{N_L}$ (whose labels are revealed). As for the unlabeled pool, we only have access to the features: $\mathcal{S}_U = \{\mathbf{x}\}^{N_U}$. We further assume that the unlabeled pool is sufficiently large so that active sampling will not change the pool distribution since only limited labels will be revealed. That is, $p(\mathbf{x}, y)$ remains fixed during AL. Our primary task couples two objectives: AL and model evaluation at the same time. In this setting, the goal of AL is to

create a labeled set by selecting samples to label and achieve the best performance. The performance of the model is indicated by the expected error when generalized to the underlying data distribution, which is often noted as the *risk R*. As discussed earlier, existing works on label-efficient model evaluation assume that the model being evaluated is already well-trained and kept fixed. They are inadequate to handle a dynamic active learning scenario, which is the focus of ATL. Given some loss function as the risk measurement $\mathcal{L}_\theta : f_\theta(\mathbf{x}) \times y \to \mathbb{R}$, our task of testing the model reduces to evaluating the following expectation:

$$R = \mathbb{E}_{\{\mathbf{x},y\}\sim\mathcal{D}}[\mathcal{L}_\theta] = \int \mathcal{L}(f_\theta(\mathbf{x}), y)p(\mathbf{x}, y)\mathrm{d}(\mathbf{x}, y) \tag{1}$$

However, directly evaluating (1) is infeasible as the true density function $p(\mathbf{x}, y)$ is unknown. So empirically, we have to adopt an approximation. In the pool-based learning setting, our approximation resorts to the expected empirical risk over the pool distribution $\mathcal{D}_{pool}$.

## 3.1 Overview: Active Risk Estimation During AL

In each AL iteration, we select to label $\mathbf{x}^*$ based on a sampling criterion $p_{sample}(\mathbf{x})$ to improve the model. We design the framework to be AL-agnostic, meaning that most standard AL sampling strategies can be applied here without further assumptions. Normally, the evaluation of the AL model is performed on a hold-out test set either at the end or during different *to-learn* data selection rounds of AL. However, the labeling cost of the test set is often times ignored. To achieve a more efficient evaluation framework, we propose to have the active risk estimation (*i.e.,* active testing) process intertwined with AL. Specifically, we use time stamps $t$ to denote each stage where we perform active testing. At time $t$, we use testing proposal $q_t(\mathbf{x})$ to select samples, $\mathcal{Q}_t = \{\mathbf{x}_t^{(1)}, ..., \mathbf{x}_t^{(n_t)}\}$, then label them to evaluate the model. We name this novel evaluation style *active quiz*. *Active quiz* is motivated by the dynamic nature of the AL process, which allows us to have an instant evaluation of the current model during AL. We also show that the quizzes can further be conveniently combined into a *final exam* for a comprehensive evaluation. In this way, we essentially divide a given budget for evaluation into small batches, which makes it possible to stop the evaluation before reaching the total budget by leveraging the signal from past quizzes.

The purpose of having the active quiz is to achieve a good estimate of the current model performance, which is not realizable in most existing AL frameworks. We use the risk $R$ to denote the performance. $R$ is dependent on the specific model, thus at each step we are trying to estimate the true risk of the model at $t$: $R(f_t(\cdot))$ [2]. Using $\mathcal{Q}_t$, we can have an estimate $\widehat{R}_{\mathcal{Q}_t}(f_t)$. We name the evaluation after the last quiz $\mathcal{Q}_T$ as the *final exam* because it includes all the quiz samples: $\{\mathcal{Q}_1, ..., \mathcal{Q}_T\}$. We use $\widetilde{R}_{\{\mathcal{Q}_1,...,\mathcal{Q}_T\}}(f_T(\cdot))$ to denote the final performance of active learning model. Note that $\widetilde{R}$ is different from each $\widehat{R}_t$ because: (i) the evaluation sets contain all the previous quiz data, and (ii) it only evaluates the final model, $f_T$. Intuitively, using $\{\mathcal{Q}_1, ..., \mathcal{Q}_T\}$ should be more preferable than using a single $\mathcal{Q}_t$ because more data samples are available to better represent the data distribution. However, the combination is not straightforward because each quiz is sampled using the optimal selection proposal at that time. We propose a principled way to combine multiple quizzes to achieve an unbiased $\widetilde{R}$ with theoretical guarantees. We further formulate an active feedback strategy to integrate the dual objectives of learning and testing and provide a theoretical underpinning to utilize some labeled test samples to improve the model's generalization capability.

## 3.2 Unbiased Low-variance Estimation of Model Risk

For a passive learning setting, where the model is fixed, the risk can be directly estimated through an importance weighting sampling procedure, such as the ARE scheme developed in [20]. Let $q(\mathbf{x})$ denote the test sample selection proposal, then the risk estimate can be defined as

$$\widehat{R}_{n,q} = \frac{1}{\sum_{i=1}^n \frac{p(\mathbf{x}^{(i)})}{q(\mathbf{x}^{(i)})}} \sum_{i=1}^n \frac{p(\mathbf{x}^{(i)})}{q(\mathbf{x}^{(i)})} \mathcal{L}(f_\theta(\mathbf{x}^{(i)}), y^{(i)}) \tag{2}$$

We can use the central limit theorem to show that $\widehat{R}_{n,q}$ is an unbiased estimation of $R$. Furthermore, their difference follows a zero-mean Gaussian asymptotically:

$$\sqrt{n}(\widehat{R}_{n,q} - R) \xrightarrow{n\to\infty} \mathcal{N}(0, \sigma_q^2), \ \sigma_q^2 = \int \frac{p(\mathbf{x})}{q(\mathbf{x})} \left( \int [\mathcal{L}(f_\theta(\mathbf{x}), y) - R]^2 p(y|\mathbf{x})dy \right) p(\mathbf{x})\mathrm{d}\mathbf{x} \tag{3}$$

---

[2]Here we use $f_t(\cdot)$ to represent $f_{\theta_t}(\cdot)$ to keep the notation uncluttered.

When $n$ is large, we also know that $n\mathrm{Var}[\widehat{R}_{n,q}] \xrightarrow{n\to\infty} \sigma_q^2$. Then, the optimal $q(\mathbf{x})$ minimizes the variance of the estimate. Using variational analysis, the optimal $q(\mathbf{x})$ is

$$q^*(\mathbf{x}) \propto p(\mathbf{x})\sqrt{\int [\mathcal{L}(f_\theta(\mathbf{x}),y) - R]^2 p(y|\mathbf{x})\mathrm{d}y} \tag{4}$$

which minimizes the expected squared difference between the estimate and the true risk (also the variance of the asymptotic Gaussian). Since the pool distribution remains fixed when only a small number of instances are sampled for AL, we can keep $p(\mathbf{x})$ as a uniform distribution over the pool. Since the importance weighting proposals are known up to a normalization factor, we have $p(\mathbf{x}) = \frac{1}{N_U}$ where $N_U$ is the pool size. To optimize model evaluation in an active learning setting, we can evaluate the model at each time stamp t in the following manner: First, we obtain an active testing proposal, denoted as $q_t$, by substituting (4) with $f_{\theta_t}$. Next, we obtain the active quizzes denoted as $\mathcal{Q}_t \sim q_t$. We then evaluate the model's performance at time t using (2).

However, there are two main limitations to this approach. First, the test proposal given by (4) contains an unknown quantity, the true risk $R$. Previous studies have attempted to address this issue by predicting the true risk from the pool. However, in section 3.3, we demonstrate that commonly used true risk predictors are not accurate, particularly in the early stages of active learning. To address this, we propose a multi-source true risk predictor. Second, the current test setting does not allow for sharing of data between previous and current quizzes due to the use of different active proposal distributions. This inefficiency not only wastes data, but also reduces the stability of the model evaluation, as the test samples in each quiz may be too small to provide a robust evaluation. Therefore, in section 3.4, we propose an estimator for the active learning setting that can integrate quizzes sampled from different proposal distributions while remaining optimal.

### 3.3 Intermediate Estimate of the True Risk

In the optimal test sample selection proposal given by (4), a critical component is the true risk, which is unknown and needs to be estimated. A straightforward solution is to leverage the current model [20] or some proxy model [14] to predict the potential risk over the entire pool as $R_\theta$. However, the following analysis shows that this is equivalent to the model uncertainty in the classification setting.

**Proposition 1.** *Under the classification setting and when a standard (i) $0-1$ loss or (ii) cross-entropy loss is used, $R_\theta$ is equivalent to measuring the model uncertainty.*

More details of the proposition are provided in Appendix B.2. The above proposition shows that $R_\theta$ can only capture the model uncertainty instead of the true risk. When the labeled training samples are limited, it is possible that the model uncertainty can not reflect the level of true risk. We propose to combine the current training risk, the model uncertainty, and the current test risk (using $\mathcal{Q}_t$ that we have) to get an aggregated multi-source estimate of the true risk:

$$R_{\theta,t}^{\text{multi}} = \frac{|\mathcal{S}_L|R_{train} + |\mathcal{S}_U|R_\theta + n\tilde{R}_t}{|\mathcal{S}_L| + |\mathcal{S}_U| + n} \tag{5}$$

The multi-source estimate can more effectively avoid overestimating the risk when only using the testing information and underestimating the risk when only using training information. By aggregating these two sides of information, neither underfitting or overfitting of the model will harm the estimation to a significant extent. The effectiveness of the multi-source aggregated risk estimation has been demonstrated in our empirical evaluation.

### 3.4 Active Quiz Integration for Final Risk Estimation

We have shown the optimal way to select test samples to evaluate a fixed model in the passive learning setting. However, we still need to extend to our scenario where there are multiple stages and AL while AT are intertwined. We propose to combine the quiz results by assigning a set of weights $\mathbf{v}_t$ on them. We will show that the proposed $\mathbf{v}_t$ is optimal given the fixed quizzes. Let's denote a sequence of independent risk estimates (treated as random variables) as $\widehat{\mathbf{R}} = (\widehat{R}_{\mathcal{Q}_1}(f_T), .. \widehat{R}_{\mathcal{Q}_T}(f_T)^\top)$, where each $\widehat{R}_{\mathcal{Q}_t}(f_T)$ asymptotically converges to the true risk $R(f_T)$. We should also note that we are using each $\widehat{R}_{\mathcal{Q}_t}$ to evaluate the same model $f_T$, not using the actual values of the estimate at time $t$ (which is the evaluation of $f_t$). That is, we always utilize previously collected sets of quiz samples to

evaluate the *current* model.

$$\widehat{R}_{\mathcal{Q}_t}(f_T) = \frac{1}{\sum_{i=1}^{n_t} w_i} \sum_{i=1}^{n_t} w_i l_i = \frac{1}{\sum_{i=1}^{n_t} \frac{p(\mathbf{x}^{(i)})}{q_t(\mathbf{x}^{(i)})}} \sum_{i=1}^{n_t} \frac{p(\mathbf{x}^{(i)})}{q_t(\mathbf{x}^{(i)})} \mathcal{L}(f_T(\mathbf{x}^{(i)}), y^{(i)})$$

$$\text{where } \sqrt{n_t}(\widehat{R}_{\mathcal{Q}_t}(f_T) - R(f_T)) \xrightarrow{n_t \to \infty} \mathcal{N}(0, \sigma_t^2(f_T)) \tag{6}$$

The variance $\sigma_t^2(f_T)$ is given by

$$\int \frac{p(\mathbf{x})}{q_t(\mathbf{x})} \left( \int [\mathcal{L}(f_T(\mathbf{x}), y) - R]^2 p(y|\mathbf{x}) \mathrm{d}y \right) p(\mathbf{x}) \mathrm{d}\mathbf{x} \tag{7}$$

We have defined the asymptotic Gaussian variance term when we use the quiz set $\mathcal{Q}_t$ from time $t$ to evaluate the final model $f_T$. Let $C_t = 1/\sigma_t^2(f_T)$ denote the model confidence of $f_T$ when evaluated on a test set that follows $q_t$. Next, we formally show that if the model $f_T$ is more confident about $q_t$, then $\mathcal{Q}_t$ should take a larger weight in the integrated final evaluation result.

**Theorem 1.** *Given fixed $\{\mathcal{Q}_1, ..., \mathcal{Q}_T\}$, the weighted combination $\tilde{R} = \sum_{t=1}^T v_t \widehat{R}_{\mathcal{Q}_t}$ where $v_t = \frac{C_t}{\sum_{t=1}^T C_t}$ is the optimal estimator constructed by all samples in $\{\mathcal{Q}_1, ..., \mathcal{Q}_T\}$.*

**Remark.** The weight $v_t$ depends on how accurate the estimate using each quiz set is. If we assume that the model becomes more accurate throughout the entire process, then we know that $\widetilde{R}_{T-1}$ should be closer to $R$ than $\widetilde{R}_{T-2}$, thus the weight on the most recent one is the largest. However, we still do not know $R$ and have to estimate the weight. Here we adopt a similar way as estimating $R_\theta$ to estimate the difference terms $C_t$ using the expected loss and $\widehat{R}_{\mathcal{Q}_t}(f_T)$: $\frac{1}{C_t} = \frac{1}{n_t} \sum_{i=1}^{n_t} \sum_y \frac{p(\mathbf{x}^{(i)})}{q_t(\mathbf{x}^{(i)})} [\mathcal{L}(f_T(\mathbf{x}^{(i)}), y) - \widehat{R}_{\mathcal{Q}_t}(f_T)]^2 p(y|\mathbf{x}^{(i)})$. With the optimal $v_t^*$, we define the final estimate result as $\widetilde{R} = \sum_{t=1}^T v_t^* \times \widehat{R}_{\mathcal{Q}_t}$. In practice, if we do not pre-set the length of the entire AL process, we can treat each stage as the final exam. After each AL sampling round, we perform an active testing round as a quiz. Then we use the integration method above to get the final estimate. This way, we will always have the optimal evaluation of the model given all the test samples that have been selected.

### 3.5 Active Feedback: Improve Model Learning Without Sacrificing Testing Accuracy

The proposed final risk estimation can provide label-efficient evaluation of the model performance. We can further improve the label efficiency on the the AL side since some of these labels can also be used for model training. Assume at the $t$-th quiz, we have sampled and labeled a set of test data $\mathcal{Q}_t$ according to the test proposal $q_t(\cdot)$. Since $\mathcal{Q}_t$ is also a labeled dataset, the active learner could leverage it as a source of feedback information to facilitate learning. In the proposed active learning-testing setting, the number of labeled samples for learning/training $N_L$ and testing $N_T$ both contribute to the limited overall annotation budget. Since learning and testing need to be sample-efficient, it is reasonable to assume that $N_L$ and $N_T$ are of similar magnitudes instead of having $N_L \gg N_T$ as in the existing active model evaluation works. We further denote $\mathcal{S}_{\text{FB}}$ as the feedback set, which contains samples from the collection of quizzes. The active feedback process can be formally defined as a subset selection problem that optimizes a joint objective of learning and testing.

$$\mathcal{S}_{\text{FB}}^* = \arg\min_{\mathcal{S}_{\text{FB}} \in \{Q_1, ..., Q_T\}} [R(f_{\theta|(\mathcal{S}_L \cup \mathcal{S}_{\text{FB}})}) + C||R - \tilde{R}_{(\{Q_1, ..., Q_T\} \setminus \mathcal{S}_{\text{FB}})}||] \tag{8}$$

where $C$ is a parameter to balance the two objectives: $(I) = R(f_{\theta|(\mathcal{S}_L \cup \mathcal{S}_{\text{FB}})})$ for learning and $(II) = ||R - \tilde{R}_{(\{Q_1, ..., Q_T\} \setminus \mathcal{S}_{\text{FB}})}||$ for testing.

**Theorem 2.** *Both the learning and testing tasks have the same label complexity, which is in the order of $\mathcal{O}(1/\sqrt{n})$. As a result, (1) the joint objective has an overall label complexity of $(\mathcal{O}(1/\sqrt{N_L + N_{FB}}) + \mathcal{O}(1/\sqrt{N_T - N_{FB}}))$; (2) the balancing parameter $C$ is in the order $\mathcal{O}(1)$.*

**Remark.** The Theorem reveals that it is possible to achieve an optimal balance between $(I)$ and $(II)$ by choosing a suitable $\mathcal{S}_{\text{FB}}$ such that we can further improve the model learning performance while maintaining the risk estimation quality from our quizzes-testing process. It provides a foundation to justify the benefit of active feedback. It is worth to note that the optimal subset selection problem is usually NP-hard, but we can still draw a useful conclusion that an appropriate subset $\mathcal{S}_{\text{FB}}$ can potentially improve the combined learning-testing objective rather than having $\mathcal{S}_L$ and $\mathcal{S}_T = \{Q_1, ..., Q_T\}$

always separated. The theorem can also be more intuitively interpreted by drawing analogy with human learning, who automatically incorporate knowledge or partial knowledge at inference time. In contrast, a typical supervised-learned ML model is frozen at evaluation/testing time. Thus, we need to explicitly add feedback so that the model can effectively improve the learning performance by getting new training samples.

Following the above theoretical result, we propose a selection proposal $q_{\mathrm{FB}}(\cdot)$ for $\mathcal{S}_{\mathrm{FB}}$ based on the following learning-testing intuitions. In standard AL, we should select samples that are considered to be the most informative and diverse ones. From the informativeness perspective, samples with high losses could be challenging for the current model. From the diversity perspective, the newly added samples should be different from the current labeled training set, which can be achieved using a diversity metric, such as $d(\mathcal{S}_L, \mathbf{x}) = \sqrt{\mathbf{x}^\top A_L^{-1} \mathbf{x}}$, where $A_L = \epsilon \mathbb{I} + \sum_{\mathbf{z} \in \mathcal{S}_L} \mathbf{z}\mathbf{z}^\top$ [11], where $\epsilon > 0$. From the evaluation perspective, removal of feedback samples should have as little impact to the risk estimation as possible. Since the test samples are chosen according to $q(\mathbf{x})$, it is less harmful to choose a feedback sample with a large $q(\mathbf{x})$ as similar testing samples are likely to be selected into the test data. To this end, we propose the following feedback selection strategy:

$$\mathbf{x}^* = \underset{(\mathbf{x},y) \in \mathcal{Q}_t}{\arg\max}\, q_{\mathrm{FB}}(\mathbf{x}, y; \eta) = \underset{(\mathbf{x},y) \in \mathcal{Q}_t}{\arg\max}\, q^{(t)}(\mathbf{x})\mathcal{L}(f_t(\mathbf{x}), y) + \eta d(\mathcal{S}_L, \mathbf{x}) \qquad (9)$$

where $\eta$ is a scaling parameter. We construct $\mathcal{S}_{\mathrm{FB}}$ after each active quiz round. The samples are added in $\mathcal{S}_L$ to re-train the model, and the risk estimation is also re-weighted after the removal.

### 3.6 Early Stopping with Instant Risk Estimations

In the ATL framework, the unbiased risk estimation after each AL step can be used to construct an effective convergence indicator to support early stopping of AL. In particular, we propose to use the change of a moving average of active risk estimations:

$$\Delta \tilde{R}_t = \frac{\sum_{i=t-w}^{t} v_i \tilde{R}_i}{\sum_{i=t-w}^{t} v_i} - \frac{\sum_{i=t-w-1}^{t-1} v_i \tilde{R}_i}{\sum_{i=t-w-1}^{t-1} v_i} \qquad (10)$$

Prior works show that leveraging the unlabeled data works well empirically to stop AL early [2, 6]. To this end, we propose to further augment $\Delta \tilde{R}_t$ with stabilized predictions (SP) from the unlabeled data [6], where $\mathrm{SP} = 1 - \mathrm{mean}_{\mathbf{x} \in \mathcal{U}}(\Delta y(\mathbf{x}))$.

## 4 Experiments

The proposed ATL framework is generic and can be applied to any existing active learners. In our experiments, we choose two commonly used models, Gaussian Processes (GP) and neural networks, to demonstrate the effectiveness of the proposed work. The GP is applied to a 2D synthetic dataset to help us understand the important behavior of the proposed test sampling strategy and how active feedback benefits the entire learning process. The neural network model is applied to relatively larger scale image datasets, including MNIST, FashionMNIST and CIFAR10 to demonstrate the practical performance of the proposed framework. As for the AL strategy, ATL is designed to be AL-agnostic. To demonstrate the general applicability, the main results are obtained using uncertainty (*i.e.,* entropy) based sampling as the AL strategy given its great popularity in many AL models. In Appendix C.2.3, we show that ATL can be easily integrated with a wide range of commonly used AL algorithms.

### 4.1 Experimental Settings

For synthetic experiments, we generate 2,500 data samples based on a moon-shaped distribution with two smaller inverted moon-shaped clusters at two remote corners. The dataset is shown in Figure 1 (a). We add the smaller clusters because the learning performance is usually closely related to how AL explores the unknown regions in the feature space and the designed shape allows us to visualize testing and feedback samples to evaluate their effectiveness. Moreover, we create two levels of imbalance in the dataset, the intra-class imbalance and the inter-class imbalance, to simulate an imbalanced class distribution. For the real-world experiments, we use the cross-entropy loss for risk evaluation. When we compare with the true risk $R$, we use the average evaluation results of the model on a large hold-out subset of the dataset (10,000 data samples) to represent $R$. The hold-out test set does not interact with the AL or AT processes, thus is considered a fair evaluation. In real-world experiments, we adopt the same procedure with the total pool containing 30,000 data samples.

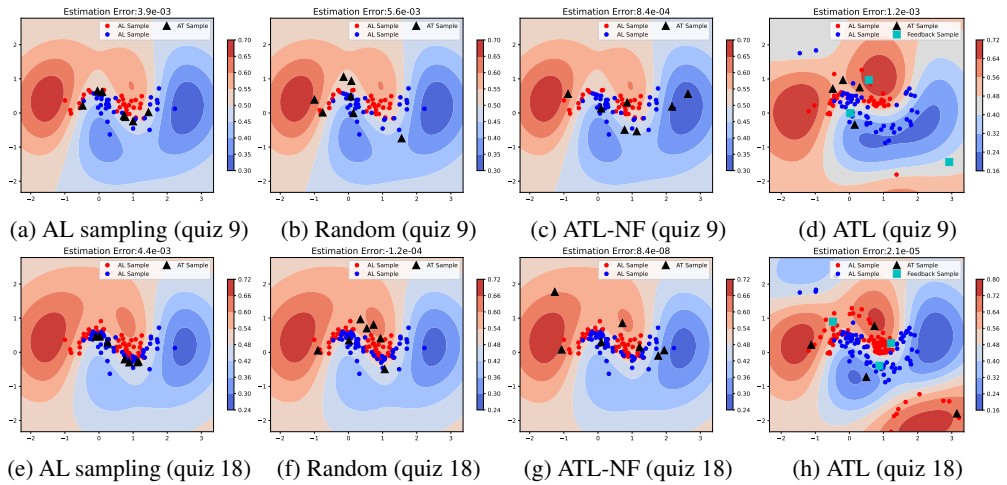

Figure 3: Active test samples selected by different criteria at quiz 9 [(a)-(d)], and 18 (e)-(h)

In each test sampling round, we compute over the entire unlabeled pool according to (4). We use a deterministic way of summing over all classes based on the posterior distribution predicted by the model. The multi-source estimate only utilizes labels of the training samples and previously selected test samples along with the predictions for unlabeled samples. Later in the active feedback stage, if a test sample is selected to be added to the training set, we remove it from the testing set. For test and feedback selections, we sequentially sample $n_t$ times to obtain each batch. The testing process does not involve re-training the model, thus the batch mode has no effect.

### 4.2 Synthetic Data Experiments

In Figure 3, we show the distribution of test samples selected using (1) AL sampling, (2) random sampling, (3) ATL w/o feedback, and (4) ATL at an early stage (*i.e.,* quiz 9) and later stage (*i.e.,* quiz 18) of the learning process. We observe that using AL sampling for test point selection has less accurate risk estimation performance due to highly biased sample selection near the decision boundary. Random sampling, on the other hand, follows the true data distribution, so it mainly selects from the central region, where the data is dense. Our proposed test sampling focuses on the most representative regions as random sampling does, but it also covers other interesting regions with relatively low density (*e.g.,* the right edge where the minority blue class is mainly located), and the model evaluation reaches the highest accuracy in the early phases of learning.

In Figure 3d, we plot the feedback with the same annotation cost to demonstrate its learning-testing trade-off. By providing part of the testing data as effective feedback, the estimation error is slightly increased. However, the sacrifice in model evaluation accuracy pays off from the training side. The feedback helps the AL model build a more representative training set and learn a more reasonable decision boundary than without feedback. It allows the AL model to gain a greater advantage in the early stage of training. Figures 3e-3h show the test data distribution in a later stage at quiz 18. The proposed testing selection is the only one that picks samples to test the critical but low-density regions. More importantly, Figure 3h shows that learning with feedback is most successful as it acquires an informative training dataset and a near-optimal decision boundary in the end.

### 4.3 Real-Data Experiments

The main results of real-world experiments consist of three parts: (1) risk estimation during standard AL (without active feedback), (2) AL performance and risk estimation with active feedback, and (3) AT-based early stopping. In this section, we focus on showing the results from the first two and leave the details of part (3) in the Appendix. In the AL without feedback case, we compare the estimation error results of ATL-NF (no feedback) with ARE quizzes integrate [20] and the adapted Active Testing [14] and Active Surrogate Estimation [15] (referred to as AT integrate and ASE integrate). We adapt the AT and ASE baseline to use in the AL setting by adjusting the pool size $N$ at each round and including the selected training loss to maintain the unbiased estimator $\tilde{R}_{LURE}$. We also need to train a surrogate model using fewer samples than normal (we use a separate NN model re-trained every few rounds). Because the AL model and the surrogate models are both severely under-trained in the early stage of AL, and that the training samples must also be considered in $\tilde{R}_{LURE}$, the estimation

error is usually significant in the AL setting. In the active feedback experiments, we compare the proposed ATL with ATL-NF and ATL-RF (random feedback).

Table 1: Estimation error: squared difference between estimate and true risks ($\times 10^{-3}$)

| Dataset | AL round / Method | 4 | 8 | 12 | 16 | 20 |
|---|---|---|---|---|---|---|
| MNIST | ARE quiz | $5.27 \pm 5.42$ | $6.39 \pm 1.54$ | $2.96 \pm 3.45$ | $8.85 \pm 4.31$ | $8.31 \pm 3.96$ |
| | AT integrate | $16.3 \pm 24.5$ | $32.8 \pm 22.1$ | $6.93 \pm 18.0$ | $8.72 \pm 3.59$ | $3.11 \pm 2.98$ |
| | ASE integrate | $3.45 \pm 2.76$ | $1.45 \pm 1.00$ | $2.17 \pm 5.06$ | $4.00 \pm 2.37$ | $5.88 \pm 5.27$ |
| | ATL-NF | $\mathbf{2.57 \pm 1.17}$ | $\mathbf{0.79 \pm 1.15}$ | $\mathbf{0.17 \pm 0.15}$ | $\mathbf{0.56 \pm 0.30}$ | $\mathbf{1.32 \pm 0.37}$ |
| Fashion MNIST | ARE quiz | $4.24 \pm 3.01$ | $4.62 \pm 7.77$ | $8.63 \pm 2.47$ | $5.71 \pm 1.87$ | $23.78 \pm 1.75$ |
| | AT integrate | $11.9 \pm 6.1$ | $36.1 \pm 30.7$ | $34.1 \pm 31.4$ | $28.0 \pm 36.9$ | $22.5 \pm 25.7$ |
| | ASE integrate | $11.1 \pm 3.63$ | $3.72 \pm 3.53$ | $3.56 \pm 8.78$ | $5.29 \pm 9.78$ | $8.42 \pm 6.72$ |
| | ATL-NF | $\mathbf{3.64 \pm 1.61}$ | $\mathbf{0.67 \pm 0.38}$ | $\mathbf{0.96 \pm 0.16}$ | $\mathbf{0.98 \pm 0.43}$ | $\mathbf{3.04 \pm 1.37}$ |
| CIFAR10 | ARE quiz | $10.1 \pm 8.79$ | $13.8 \pm 13.0$ | $22.2 \pm 14.7$ | $21.9 \pm 31.4$ | $14.1 \pm 13.4$ |
| | AT integrate | $6.89 \pm 6.98$ | $12.0 \pm 7.18$ | $21.8 \pm 5.73$ | $12.9 \pm 9.76$ | $38.9 \pm 25.6$ |
| | ASE integrate | $10.9 \pm 3.67$ | $6.51 \pm 2.87$ | $7.53 \pm 1.46$ | $17.6 \pm 2.66$ | $23.2 \pm 6.10$ |
| | ATL-NF | $\mathbf{8.83 \pm 7.79}$ | $\mathbf{3.06 \pm 5.04}$ | $\mathbf{4.95 \pm 7.12}$ | $\mathbf{7.94 \pm 5.22}$ | $\mathbf{6.20 \pm 5.79}$ |

Table 2: Hold-out test risk using different feedback criteria over 20 AL rounds

| Dataset | AL round / Method | 4 | 8 | 12 | 16 | 20 |
|---|---|---|---|---|---|---|
| MNIST | ATL-NF | $0.92 \pm 0.06$ | $0.55 \pm 0.08$ | $0.46 \pm 0.06$ | $0.32 \pm 0.04$ | $0.22 \pm 0.02$ |
| | ATL-RF | $0.92 \pm 0.12$ | $0.54 \pm 0.02$ | $0.41 \pm 0.05$ | $0.29 \pm 0.03$ | $0.21 \pm 0.02$ |
| | ATL | $\mathbf{0.88 \pm 0.07}$ | $\mathbf{0.53 \pm 0.04}$ | $\mathbf{0.39 \pm 0.03}$ | $\mathbf{0.26 \pm 0.01}$ | $\mathbf{0.19 \pm 0.03}$ |
| Fashion MNIST | ATL-NF | $0.75 \pm 0.03$ | $0.69 \pm 0.02$ | $0.61 \pm 0.02$ | $0.57 \pm 0.04$ | $0.56 \pm 0.03$ |
| | ATL-RF | $0.75 \pm 0.04$ | $0.68 \pm 0.02$ | $0.61 \pm 0.01$ | $0.58 \pm 0.06$ | $0.56 \pm 0.04$ |
| | ATL | $\mathbf{0.74 \pm 0.03}$ | $\mathbf{0.65 \pm 0.04}$ | $\mathbf{0.59 \pm 0.02}$ | $\mathbf{0.56 \pm 0.03}$ | $\mathbf{0.51 \pm 0.01}$ |
| CIFAR10 | ATL-NF | $1.91 \pm 0.04$ | $1.76 \pm 0.05$ | $1.72 \pm 0.01$ | $1.66 \pm 0.02$ | $1.55 \pm 0.03$ |
| | ATL-RF | $1.91 \pm 0.03$ | $1.77 \pm 0.04$ | $1.69 \pm 0.03$ | $1.60 \pm 0.04$ | $1.54 \pm 0.07$ |
| | ATL | $\mathbf{1.90 \pm 0.05}$ | $\mathbf{1.76 \pm 0.02}$ | $\mathbf{1.65 \pm 0.03}$ | $\mathbf{1.58 \pm 0.02}$ | $\mathbf{1.53 \pm 0.02}$ |

Table 3: Estimation error with feedback over 20 AL rounds ($\times 10^{-3}$)

| Dataset | AL round / Method | 4 | 8 | 12 | 16 | 20 |
|---|---|---|---|---|---|---|
| MNIST | ATL-RF | $26.8 \pm 21.4$ | $21.4 \pm 17.0$ | $3.54 \pm 4.01$ | $5.54 \pm 3.21$ | $7.62 \pm 4.41$ |
| | ATL | $\mathbf{14.6 \pm 22.1}$ | $\mathbf{16.9 \pm 13.7}$ | $\mathbf{3.19 \pm 2.63}$ | $\mathbf{4.15 \pm 3.20}$ | $\mathbf{1.87 \pm 1.41}$ |
| Fashion MNIST | ATL-RF | $10.2 \pm 9.30$ | $4.41 \pm 3.77$ | $2.19 \pm 5.53$ | $5.69 \pm 4.52$ | $11.6 \pm 7.51$ |
| | ATL | $\mathbf{2.50 \pm 2.93}$ | $\mathbf{1.94 \pm 2.25}$ | $\mathbf{1.78 \pm 1.07}$ | $\mathbf{6.32 \pm 5.41}$ | $\mathbf{5.03 \pm 4.41}$ |
| CIFAR10 | ATL-RF | $20.6 \pm 17.6$ | $19.1 \pm 13.7$ | $9.82 \pm 8.03$ | $33.6 \pm 30.5$ | $24.8 \pm 32.4$ |
| | ATL | $\mathbf{11.6 \pm 13.4}$ | $\mathbf{5.11 \pm 3.45}$ | $\mathbf{8.81 \pm 6.51}$ | $\mathbf{11.9 \pm 16.7}$ | $\mathbf{6.57 \pm 6.29}$ |

**AL-AT results.** We use squared difference between the estimated risk and the true risk as well as the variance of the squared difference to measure the estimation error instead of the variance of the estimate itself because each time we repeat the experiment the model is different. This is also a difference caused by studying the AL process, instead of evaluating a fixed model. From Table 1, we see that ATL-NF has a much lower error of estimation than single ARE quizzes and AT/ASE integrate. The ARE quizzes put all testing samples without adjusting the importance weights, thus will result in a bigger estimation error. The drawback of the AT method is that the surrogate model is not a very reliable reference and the evaluated model changes over time. Because it selects samples based on a max-score criterion instead of importance sampling, the change of model may affect the unbiased estimate. The ASE integrate has the same issue of relying on a good surrogate model. However, the risk estimation of ASE is also based on the surrogate model predictions, which is consistent with the test sample selection. Thus, it outperforms AT integrate most of the time.

**Active feedback results.** We compare traditional AL and active learning-testing with feedback. We use both the cross-entropy loss as the metrics for the model performance. In Table 2, we see that the proposed feedback and random feedback can both help the model reach a smaller testing loss than pure AL with the same total number of labels. The initial training set contains 500 labels, while each of the 20 AL rounds adds 500 labels. AT then samples 100 points after each AL round, and the active feedback sends 50 labels back to training. Because of the importance sampling with replacement nature of the process, the feedback approaches actually use fewer total labels in the end. We also study the impact of feedback on the estimation error in Table 3. In the early stage of AL, it is obvious that random feedback is harmful to the estimation error. The proposed approach also suffers from the increase of the estimation error. However, as the AL goes on, our proposed feedback selection can maintain a low level of estimation error, while improving the model performance. Combining these

benefits, we can potentially save the labeling budget for both learning and evaluation purposes. We report further ablation studies on the feedback proposal in Appendix C.

We also provide a study on the size of the feedback set. As mentioned in the proof for Theorem 2, we keep the size of feedback simple in this work. This is to be consistent with our theoretical analysis and the experiments show that the active feedback process is helpful in this generic setting. Further details about extending this have been discussed as part of the future directions. However, even in the simple setting of fixed feedback size, we can see that the learning and testing performances do not consistently and monotonically change with respect to the feedback size. From Tables 4 and 5 below, we can see that in general, model risk (learning performance) is better when we use a larger feedback size, but at the same time the estimation error (testing performance) may become much worse. The model risk on CIFAR10 behaves differently with the feedback size, probably because the model performance is not good enough and adding difficult samples in this stage does not necessarily help with the generalization ability.

Table 4: Hold-out test risk using different feedback criteria over 20 AL rounds

| Dataset | Feedback size / AL round | 4 | 8 | 12 | 16 | 20 |
|---|---|---|---|---|---|---|
| MNIST | 83% | $0.86 \pm 0.09$ | $0.53 \pm 0.04$ | $0.40 \pm 0.08$ | $0.30 \pm 0.02$ | $0.20 \pm 0.03$ |
| | 67% | $0.87 \pm 0.08$ | $0.52 \pm 0.03$ | $0.35 \pm 0.03$ | $0.30 \pm 0.02$ | $0.21 \pm 0.02$ |
| | 50% | $0.88 \pm 0.07$ | $0.53 \pm 0.04$ | $0.39 \pm 0.03$ | $0.26 \pm 0.01$ | $0.19 \pm 0.03$ |
| | 25% | $0.94 \pm 0.06$ | $0.54 \pm 0.03$ | $0.42 \pm 0.08$ | $0.35 \pm 0.02$ | $0.25 \pm 0.02$ |
| | 20% | $0.99 \pm 0.04$ | $0.56 \pm 0.08$ | $0.43 \pm 0.06$ | $0.38 \pm 0.01$ | $0.24 \pm 0.02$ |
| Fashion MNIST | 83% | $0.74 \pm 0.02$ | $0.67 \pm 0.03$ | $0.60 \pm 0.03$ | $0.54 \pm 0.02$ | $0.51 \pm 0.03$ |
| | 67% | $0.77 \pm 0.04$ | $0.68 \pm 0.03$ | $0.59 \pm 0.03$ | $0.56 \pm 0.03$ | $0.52 \pm 0.02$ |
| | 50% | $0.74 \pm 0.03$ | $0.65 \pm 0.04$ | $0.59 \pm 0.02$ | $0.56 \pm 0.03$ | $0.51 \pm 0.01$ |
| | 25% | $0.76 \pm 0.02$ | $0.70 \pm 0.01$ | $0.62 \pm 0.02$ | $0.59 \pm 0.05$ | $0.53 \pm 0.03$ |
| | 20% | $0.77 \pm 0.02$ | $0.71 \pm 0.02$ | $0.64 \pm 0.02$ | $0.61 \pm 0.04$ | $0.54 \pm 0.04$ |
| CIFAR10 | 83% | $1.92 \pm 0.06$ | $1.71 \pm 0.02$ | $1.67 \pm 0.07$ | $1.59 \pm 0.04$ | $1.57 \pm 0.04$ |
| | 67% | $1.96 \pm 0.05$ | $1.75 \pm 0.02$ | $1.64 \pm 0.04$ | $1.58 \pm 0.04$ | $1.58 \pm 0.06$ |
| | 50% | $1.90 \pm 0.05$ | $1.76 \pm 0.02$ | $1.65 \pm 0.03$ | $1.58 \pm 0.02$ | $1.53 \pm 0.02$ |
| | 25% | $1.94 \pm 0.08$ | $1.76 \pm 0.03$ | $1.70 \pm 0.03$ | $1.64 \pm 0.04$ | $1.59 \pm 0.02$ |
| | 20% | $1.91 \pm 0.03$ | $1.76 \pm 0.02$ | $1.73 \pm 0.03$ | $1.59 \pm 0.02$ | $1.63 \pm 0.02$ |

Table 5: Squared difference between the estimate and the true risk over 20 AL rounds ($\times 10^{-3}$)

| Dataset | Feedback size / AL round | 4 | 8 | 12 | 16 | 20 |
|---|---|---|---|---|---|---|
| MNIST | 83% | $50.2 \pm 39.8$ | $21.0 \pm 24.3$ | $7.36 \pm 8.44$ | $11.4 \pm 12.7$ | $7.59 \pm 4.45$ |
| | 67% | $25.6 \pm 23.4$ | $29.3 \pm 29.7$ | $6.90 \pm 8.05$ | $6.24 \pm 6.71$ | $7.50 \pm 5.07$ |
| | 50% | $14.6 \pm 22.1$ | $16.9 \pm 13.7$ | $3.19 \pm 2.63$ | $4.15 \pm 3.20$ | $1.87 \pm 1.41$ |
| | 25% | $11.7 \pm 11.5$ | $10.0 \pm 7.98$ | $9.73 \pm 11.4$ | $4.76 \pm 5.25$ | $1.59 \pm 1.96$ |
| | 20% | $28.0 \pm 24.4$ | $11.8 \pm 14.5$ | $5.91 \pm 3.82$ | $4.31 \pm 4.80$ | $1.25 \pm 1.36$ |
| Fashion MNIST | 83% | $8.39 \pm 8.97$ | $7.52 \pm 10.4$ | $2.77 \pm 3.58$ | $3.87 \pm 4.45$ | $11.1 \pm 7.02$ |
| | 67% | $8.59 \pm 8.77$ | $8.60 \pm 10.5$ | $5.42 \pm 5.96$ | $4.05 \pm 2.47$ | $14.6 \pm 13.8$ |
| | 50% | $2.50 \pm 2.93$ | $1.94 \pm 2.25$ | $1.78 \pm 1.07$ | $6.32 \pm 5.41$ | $5.03 \pm 4.41$ |
| | 25% | $3.04 \pm 4.00$ | $2.38 \pm 4.81$ | $1.54 \pm 1.18$ | $6.40 \pm 8.06$ | $4.13 \pm 3.99$ |
| | 20% | $2.62 \pm 1.57$ | $1.56 \pm 1.77$ | $2.42 \pm 4.52$ | $5.65 \pm 4.33$ | $5.22 \pm 3.27$ |
| CIFAR10 | 83% | $54.5 \pm 54.1$ | $14.3 \pm 7.75$ | $56.1 \pm 17.0$ | $47.2 \pm 34.3$ | $62.2 \pm 43.3$ |
| | 67% | $24.6 \pm 25.6$ | $36.7 \pm 20.5$ | $24.1 \pm 18.6$ | $30.7 \pm 40.8$ | $36.2 \pm 21.0$ |
| | 50% | $11.6 \pm 13.4$ | $5.11 \pm 3.45$ | $8.81 \pm 6.51$ | $11.9 \pm 16.7$ | $6.57 \pm 6.29$ |
| | 25% | $4.88 \pm 5.80$ | $6.01 \pm 8.22$ | $6.80 \pm 1.36$ | $10.2 \pm 13.4$ | $4.48 \pm 3.53$ |
| | 20% | $5.44 \pm 6.65$ | $3.65 \pm 3.44$ | $11.2 \pm 11.0$ | $4.21 \pm 1.36$ | $5.82 \pm 3.34$ |

## 5   Conclusion

We address a real-world challenge for sample-efficient learning, where valuable labels from human experts that can be used for testing/evaluation are also scarce. We propose an ATL framework that builds an active learning-testing-feedback loop to achieve a label-efficient evaluation of an AL model on the fly, with the potential to further improve the model performance without extra labels and opening up to the creation of systematic early stopping criteria. We theoretically prove that ATL has an unbiased and label-efficient estimator and provide an analysis that shows how the label complexity dependencies are maintained through active feedback. The experiments show that ATL optimizes the number of human labels needed in learning by simultaneously acting as a fair referee and an educative teacher.

## Acknowledgements

This research was supported in part by an NSF IIS award IIS-1814450 and an ONR award N00014-18-1-2875. The views and conclusions contained in this paper are those of the authors and should not be interpreted as representing any funding agency. We would also like to thank the anonymous reviewers for their constructive comments.

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

# Appendix

## Table of Contents

## Appendix Overview

**Organization.** The Appendix is organized as follows. We first provide a summary of notations in Appendix A. Then, we present the details of the theoretical analysis in the main paper in Appendix B. We further provide additional experimental results in Appendix C, including a synthetic visualization for different feedback approaches in Appendix C.1 and more detailed real-world experiments in Appendix C.2. The hardware information is presented in Appendix D. We discuss the limitation, future direction, and social impact of the proposed work in Appendix E. Th algorithm block for the pseudo code and the the link to the source code are presented in Appendix F.

## A    Summary of Notations

Table 6:  Summary of key notations with definitions

| Notation | Definition |
| --- | --- |
| $(\mathbf{x}, y) \in \mathcal{X} \times \mathcal{Y}$ | Data points |
| $D; p(\mathbf{x}, y)$ | Data distribution |
| $\mathcal{L}_\theta$ | Loss function of model $f_\theta()$ |
| $R$ | True risk |
| $R(f_{\theta|D})$ | True risk evaluated on model $f$ whose parameter $\theta$ is learned from dataset $D$. |
| $\mathcal{S}_L, \mathcal{S}_U$ | Labeled set and unlabeled pool |
| $\mathcal{Q}_t = \{\mathbf{x}_t\}^{n_t}$ | The $t$-th quiz set |
| $q(\mathbf{x}), q^*(\mathbf{x})$ | Test sample selection proposal and the optimal proposal |
| $\hat{R}_q, \hat{R}_t$ | Risk estimator indexed by the test proposal $q$ or time step $t$ |
| $\tilde{R}$ | Integrated risk estimator |
| $C_t, v_t$ | Model confidence of $f_t$ and the weight coefficient for time step $t$ in final $\tilde{R}$ |
| $\mathcal{S}_{\text{FB}}$ | Active feedback set |
| $N_L, N_T, N_{\text{FB}}$ | Number of samples in learning, testing and feedback sets |
| $d(\cdot, \cdot), A_L, \epsilon$ | Diversity metric, diversity norm matrix, small positive value $\epsilon$ to avoid singular issues |
| $q_{\text{FB}}(\mathbf{x}), \eta$ | Feedback proposal, balancing parameter between the proposal-loss term and the diversity term in the feedback proposal |
| $\lambda$ | Balancing parameter for the risk estimation in unlabeled-information-combined early stopping criterion |

## B    Proof and Additional Analysis of Main Theoretical Results

### B.1    Proof of Theorem 1

*Proof.* We start by presenting the asymptotic convergence of the active risk estimator and the solution for the optimal testing selection proposal $q^*(\mathbf{x})$. From [20], we know that using the risk estimator $\hat{R}_{n,q}$ we would get an unbiased estimate of the true risk $R$ because it is essentially an importance sampling based estimator. Then from the central limit theorem, $\hat{R}_{n,q}^0 = \sum_{i=1}^n w^{(i)} l^{(i)}$ and $W_n = \sum_{i=1}^n w^{(i)}$ are asymptotically normally distributed with

$$\sqrt{n} \left( \frac{1}{n} \hat{R}_{n,q}^0 - R \right) \xrightarrow{n \to \infty} \mathcal{N}(0, \text{var}[w^{(i)} l^{(i)}]) \tag{11}$$

$$\sqrt{n} \left( \frac{1}{n} W_n - 1 \right) \xrightarrow{n \to \infty} \mathcal{N}(0, \text{var}[w^{(i)}]) \tag{12}$$

Then, with the multivariate delta method, we know that if $Y_n = (Y_{n1}, ..., Y_{nk})$ is a sequence and $\sqrt{n}(Y_n - \mu) \xrightarrow{n \to \infty} \mathcal{N}(0, \Sigma)$, then

$$\sqrt{n}(g(Y_n) - g(\mu)) \xrightarrow{n \to \infty} \mathcal{N}(0, \nabla g(y)^\top \Sigma \nabla g(y)) \tag{13}$$

Here the function is $g(x,y) = \frac{x}{y}$ with $x = \frac{1}{n}\hat{R}^0_{n,q}$ and $y = \frac{1}{n}W_n$. The result is

$$\sqrt{n}\left(\frac{\frac{1}{n}\hat{R}^0_{n,q}}{\frac{1}{n}W_n} - R\right) \xrightarrow{n\to\infty} \mathcal{N}(0, \sigma_q^2) \tag{14}$$

where $\sigma_q^2 = \int \frac{p(\mathbf{x})}{q(\mathbf{x})}\left(\int[\mathcal{L}(f(\mathbf{x}),y) - R(f)]^2 p(y|\mathbf{x})dy\right)p(\mathbf{x})d\mathbf{x}$.

Then, the optimal test proposal is obtained by minimizing $\sigma_q^2$. By introducing a Lagrange multiplier $\beta$ for the constraint $\int q(\mathbf{x})d\mathbf{x} = 1$, we have

$$L(q,\beta) = \sigma_q^2 + \beta\left(\int q(\mathbf{x})d\mathbf{x} - 1\right) \tag{15}$$

$$\frac{\partial L}{\partial q} = -\frac{p(\mathbf{x})^2 \int[\mathcal{L}(f(\mathbf{x}),y) - R(f)]^2 p(y|\mathbf{x})dy}{q(\mathbf{x})^2} + \beta = 0 \tag{16}$$

Thus, we have $q^*(\mathbf{x}) \propto p(\mathbf{x})\sqrt{\int[\mathcal{L}(f(\mathbf{x}),y) - R(f)]^2 p(y|\mathbf{x})dy}$.

Now, we provide the detailed proof for Theorem 1. As shown in Section 3.4, $\hat{\mathbf{R}}$ satisfies

$$\sqrt{n_t}(\hat{\mathbf{R}} - R\mathbb{1}) \sim \mathcal{N}\left(\mathbf{0}, \text{diag}\left[\sigma_1^2, ..\sigma_T^2\right]^\top\right) \tag{17}$$

Next, we apply the multi-variant delta method. Define $g : \mathbb{R}^T \to \mathbb{R}$, $g(\hat{\mathbf{R}}) = \sum_{t=1}^T v_t \hat{R}_{\mathcal{Q}_t}$. Then, we have $\bigtriangledown g = (v_1, ..., v_t)^\top$. Given the diagonal covariance matrix, the final variance is:

$$\sigma_T^2 = (v_1, ..., v_t)\begin{pmatrix}\sigma_1^2 & & \\ & ... & \\ & & \sigma_T^2\end{pmatrix}(v_1, ..., v_t)^\top$$

$$= \sum_{t=1}^T \int \frac{p(\mathbf{x})}{q_t(\mathbf{x})}v_t^2\left(\int[\mathcal{L}(f_T(\mathbf{x}),y) - R(f_T)]^2 p(y|\mathbf{x})dy\right)p(\mathbf{x})d\mathbf{x} \tag{18}$$

When we perform the "final exam" estimation after gathering all quizzes $\{\mathcal{Q}_1, ..., \mathcal{Q}_T\}$, the other factors including testing proposals are fixed. We analyze the optimal solution for $v_t$ by constructing the Lagrangian objective $\sigma_T^2 + \gamma(\sum_t v_t - 1)$ (where $\gamma$ is a Lagrangian multiplier). By taking the derivative w.r.t each $v_t$ along with the Lagrangian, we have

$$\frac{\partial[\sum_{t=1}^T v_t^2(\sigma_t^2) + \gamma(v_t - 1/T)]}{\partial v_t} = 0 \tag{19}$$

which leads to $v_t = \frac{C_t}{\sum_{t=1}^T C_t}$. $\qquad\square$

The Corollary below provides an alternative view of Theorem 1.

**Corollary 1.** *If we do not change individual $q_t$ but still combine all available test samples, then adjusting their importance weight by $w'^{(i)}_t = v_t \times w^{(i)}_t$ gives the optimal estimator.*

*Proof.* In the alternative view, we have:

$$\widetilde{R} = \frac{\widetilde{R}^0}{W'} = \frac{\sum_{t=1}^T \sum_{i=1}^{n_t} v_t w^{(i)}_t l^{(i)}_t}{\sum_{t=1}^T \sum_{i=1}^{n_t} v_t w^{(i)}_t} \tag{20}$$

where $w^{(t)}_i = \frac{p(\mathbf{x}^{(i)})}{q_t(\mathbf{x}^{(i)})}$. We can view the final estimate $\widetilde{R}$ as a function of $\widetilde{R}^0$ and $W'$ that has the form $f(X,Y) = \frac{X}{Y}$. Then we directly analyze the expectation and variance of $\widetilde{R}$ using the delta method: First we have

$$\begin{aligned}
\mathbb{E}(f(X,Y)) &= \mathbb{E}[f(\mu_X, \mu_Y) + f'_Y(\mu_X, \mu_Y)(X - \mu_X) + f'_Y(\mu_X, \mu_Y)(Y - \mu_Y) + R] \\
&\approx \mathbb{E}[f(\mu_X, \mu_Y)] + \mathbb{E}[f'_X(\mu_X, \mu_Y)(X - \mu_X)] + \mathbb{E}[f'_Y(\mu_X, \mu_Y)(Y - \mu_Y)] \\
&= \mathbb{E}[f(\mu_X, \mu_Y)] + f'_X(\mu_X, \mu_Y)\mathbb{E}[(X - \mu_X)] + f'_Y(\mu_X, \mu_Y)\mathbb{E}[(Y - \mu_Y)] \\
&= f(\mu_X, \mu_Y) \tag{21}
\end{aligned}$$

where $\mu_X = \mathbb{E}[X]$ and $\mu_Y = \mathbb{E}[Y]$. Applying (21) on our estimate, and we get:

$$\mathbb{E}[\tilde{R}(f_T)] = \mathbb{E}\left[\frac{\sum_{t=1}^{T}\sum_{i=1}^{n} v_t w_t^{(i)} l_{it}}{\sum_{t=1}^{T}\sum_{i=1}^{n} v_t w_t^{(i)}}\right] = \frac{\sum_{t=1}^{T} v_t \mathbb{E}[\tilde{R}_t^0]}{\sum_{t=1}^{T} v_t W_{n,t}} = R$$

where we utilize $\sum_{t=1}^{T} v_t = 1$ and $\mathbb{E}[\frac{\tilde{R}_t^0}{W_t'}] = R$. For the variance, we have:

$$\begin{aligned}
\mathrm{Var}[f(X,Y)] &= \mathbb{E}[(f(X,Y) - \mathbb{E}[f(X,Y)])^2] \\
&\approx \mathbb{E}[(f(X,Y) - f(\mu_X,\mu_Y))^2] \\
&\approx \mathbb{E}[(f(\mu_X,\mu_Y) + f_X'(\mu_X,\mu_Y)(X-\mu_X) + f_Y'(\mu_X,\mu_Y)(Y-\mu_Y) - f(\mu_X,\mu_Y))^2] \\
&= \mathbb{E}[f_X'^2(\mu_X,\mu_Y)(X-\mu_X)^2 + 2f_X'(\mu_X,\mu_Y)(X-\mu_X)f_Y'(\mu_X,\mu_Y)(Y-\mu_Y) \\
&\quad + f_Y'^2(\mu_X,\mu_Y)(Y-\mu_Y)^2] \\
&= f_X'^2(\mu_X,\mu_Y)\mathrm{Var}[X] + 2f_X'(\mu_X,\mu_Y)f_Y'(\mu_X,\mu_Y)\mathrm{Cov}[X,Y] + f_Y'^2(\mu_X,\mu_Y)\mathrm{Var}[Y]
\end{aligned}$$
(22)

Applying to our estimate leads to

$$\begin{aligned}
\mathrm{Var}[\tilde{R}] &\approx R^2 \mathrm{Var}[W'] + \mathrm{Var}[\tilde{R}^0] - 2R\mathrm{Cov}[W', \tilde{R}^0] \\
&= R^2(\mathbb{E}[W'^2] - \mathbb{E}^2[W']) + (\mathbb{E}[(\tilde{R}^0)^2] - \mathbb{E}^2[\tilde{R}^0]) - 2R(\mathbb{E}[W'\tilde{R}^0] - \mathbb{E}[\tilde{R}^0]\mathbb{E}[W']) \\
&= R^2\mathbb{E}[W'^2] - 2R\mathbb{E}[W'\tilde{R}^0] + \mathbb{E}[(\tilde{R}^0)^2] \\
&= \sum_{t=1}^{T}\int \frac{p(\mathbf{x})}{q_t(\mathbf{x})} v_t^2 \left(\int [\mathcal{L}(f_T(\mathbf{x}),y) - R(f_T)]^2 p(y|\mathbf{x})dy\right) p(\mathbf{x})d\mathbf{x}
\end{aligned}$$
(23)

where we utilize $f(X,Y) = \frac{X}{Y} \to f_X' = \frac{1}{Y}$, $f_Y' = -\frac{X}{Y^2}$, $\mu_X = R, \mu_Y = 1$. Note that since we assume $q_t(\mathbf{x})$ are fixed, we have $\mathbb{E}[W'\tilde{R}^0] = \mathbb{E}_{p(y|\mathbf{x})}\mathbb{E}_{q_1}...\mathbb{E}_{q_T}[\sum_{t=1}^{T} v_t \sum_{i=1}^{n}(w_t^{(i)})^2 l(\mathbf{x}_t^{(t)})] = \mathbb{E}_{p(y|\mathbf{x})}[\sum_{t=1}^{T} v_t \mathbb{E}_{q_t}\sum_{i=1}^{n}(w_t^{(i)})^2 l(\mathbf{x}_t^{(t)})]$. $\quad\square$

## B.2 Proposition 1

We show two concrete examples for Proposition 1. In each case, the estimated introspective loss is analogous to an uncertainty measure.

- The estimation of 0-1 loss is:

$$R_\theta = \frac{1}{|\mathcal{S}_U|}\sum_{\mathbf{x}\in\mathcal{S}_U}\sum_y \mathbb{1}(f_\theta(\mathbf{x}) \neq y)p(y|\mathbf{x};\theta)$$
(24)

  which is the sum of the predicted probability of all classes other than the most probable class.

- The estimation of cross-entropy loss is:

$$R_\theta = \frac{1}{|\mathcal{S}_U|}\sum_{\mathbf{x}\in\mathcal{S}_U}\sum_y p(y|\mathbf{x};\theta)\log(p(y|\mathbf{x};\theta))$$
(25)

  which is the entropy of the predicted probability.

When we use deep learning models, $R_\theta$ usually largely underestimates the risk over the entire pool. In other works such as [14, 15], the surrogate risk acts in a similar way. For the final risk estimator to be accurate, the introspective risk estimation or the surrogate risk first needs to be accurate, which somewhat beats the purpose of active risk estimation. However, we still try to improve this intermediate step without assuming that we have access to an unrealistically accurate estimation, leading to our proposed $R_\theta$ in Section 3.3.

## B.3 Proof of Theorem 2 and Active Feedback Analysis

In Theorem 2, we formalize the combined learning-testing objective as a joint optimization problem with the variable being a subset $\mathcal{S}_{\text{FB}}$ that can be transferred from the testing set $\mathcal{S}_T$ to the learning set $\mathcal{S}_L$. We define the process of selecting the subset as the "active feedback" process, which connects the learning and testing objectives through a balancing parameter $C$ given in (8). Performing exact optimization of the subset along with a parameter $C$ would require more detailed knowledge on the learning model and the AL strategy. We instead provide a general analysis to show that active feedback could indeed provide an optimal solution for the joint optimization problem, where $C$ scales as $\mathcal{O}(1)$. Following our theoretical result, we empirically demonstrate the effectiveness of an intuitive feedback approach in the experimental sections (Section 4.3, Appendix C).

**Proof overview.** We apply some generic generalization bound (*e.g.,* [17] for CNN or similar models) to the learning objective (I) in the joint optimization problem given by (8), which gives $\mathcal{O}(1/\sqrt{n})$. We then leverage the confidence interval to get a high probability bound for the testing objective (II), which also gives $\mathcal{O}(1/\sqrt{n})$ [4, 9, 28]. We use the formalized results on the convergence of the estimate as introduced in [20]. With that, we continue to show that both the learning and testing objectives share the same dependency on $n$. These common dependencies on $n$ give us the foundation to further analyze the feedback process. We offer an intuitive justification of active feedback as follows. The risk estimators are importance weighted estimates of the true risk. The estimate converges to the true risk asymptotically, so fewer samples might hurt the quality of the estimate (due to a large variance), but does not change the fact that the expected average of the estimate is still the true risk. With the confidence interval conversions, we can see that except for the change of constants, the objective's dependency on the number of samples does not change. (This also provides guidance for the feedback proposal later: if we can keep the change of the estimate to the minimum, meanwhile using the samples discarded from the test set to improve the AL model as much as possible, it would be the ideal use of available labels.) Following these high-level ideas as described above, we present the detailed proof below.

*Proof.* We first break the joint (I) learning-(II) testing objective into two parts and approach each part separately:

$$R(f_{\theta|(\mathcal{S}_L \cup \mathcal{S}_{\text{FB}})}) \leq R_{CNN}(f^*_{\theta|(\mathcal{S}_L \cup \mathcal{S}_{\text{FB}})}) + \mathcal{O}\left(1/\sqrt{N_L + N_{\text{FB}}}\right)$$
$$\lesssim R_{CNN}(f^*_{\theta|(\mathcal{S}_L)}) + \mathcal{O}\left(1/\sqrt{N_L + N_{\text{FB}}}\right) \quad (26)$$

$$\|R - \tilde{R}_{(\{Q_1,...,Q_T\}\setminus\mathcal{S}_{\text{FB}})}\| \leq \|\tilde{R}_T(\{Q_1,...,Q_T\}) - \tilde{R}_T(\{Q_1,...,Q_T\}\setminus\mathcal{S}_{\text{FB}})\|$$
$$+ \|\tilde{R}_T(\{Q_1,...,Q_T\}) - R\| \quad (27)$$

**The learning objective.** As mentioned earlier, (26) is a common generalization error bound for CNN or similar models. For example, given a training set $\mathcal{S}_L$ with $N_L$ samples, we can draw from the basic bound (*e.g.,* according to Theorem 2.1 in [17]):

$$R(f_{\theta|\mathcal{S}_L}) = \mathbb{E}_{\mathcal{D}}[l_{f_{\theta|\mathcal{S}_L}}(\cdot)] \leq \mathbb{E}_{\mathcal{S}_L}[l_{f_{\theta|\mathcal{S}_L}}(\cdot)] + C'\left(\beta'\lambda'\sqrt{\frac{|\theta|}{N_L}} + \sqrt{\frac{\log(1/\delta)}{N_L}}\right) \quad (28)$$

with probability of at least $1 - \delta$, where $C'$, $\beta'$, and $\lambda'$ are constants and $|\theta|$ is the total number of trainable parameters in the network. In our case, we do not make further assumptions about the constants and $|\theta|$ is fixed for evaluating a certain model. Similarly, we can substitute $N_L$ with $N_L + N_{\text{FB}}$ and arrive at:

$$R(f_{\theta|(\mathcal{S}_L \cup \mathcal{S}_{\text{FB}})}) = \mathbb{E}_{\mathcal{D}}[l_{f_{\theta|(\mathcal{S}_L \cup \mathcal{S}_{\text{FB}})}}(\cdot)] \leq \mathbb{E}_{(\mathcal{S}_L \cup \mathcal{S}_{\text{FB}})}[l_{f_{\theta|(\mathcal{S}_L \cup \mathcal{S}_{\text{FB}})}}(\cdot)]$$
$$+ C'\left(\beta'\lambda'\sqrt{\frac{|\theta|}{N_L + N_{\text{FB}}}} + \sqrt{\frac{\log(1/\delta)}{N_L + N_{\text{FB}}}}\right) \quad (29)$$

We notice that in both (28) and (29), we include the expected loss which is slightly different from the best possible AL model risks $R(f^*_{\theta|(\mathcal{S}_L \cup \mathcal{S}_{\text{FB}})})$ and $R(f^*_{\theta|\mathcal{S}_L})$. However, the difference is usually

on a smaller scale than $(1/\sqrt{N_L} - 1/\sqrt{N_L + N_{FB}})$. In general, we assume that $R(f^*_{\theta|(\mathcal{S}_L \cup \mathcal{S}_T)}) \lesssim R(f^*_{\theta|(\mathcal{S}_L \cup \mathcal{S}_{FB})}) \lesssim R(f^*_{\theta|\mathcal{S}_L})$ since more labeled samples can benefit learning (we do not need to assume a strictly monotonic case for the sake of this analysis). For most AL strategies, the difference between the expected empirical risk and the optimal risks given the learning set size is on a higher order of dependency on $n$ than the learning bound itself. If we ignore the higher order terms, we can simplify the results as shown in (26). Then, the term and more importantly the change in the learning objective that is related to the assumed feedback $\mathcal{S}_{FB}$ is only dependent on $N_{FB}$ through $\mathcal{O}(1/\sqrt{N_L + N_{FB}})$.

**The testing objective.** The relation in (27) can be further analyzed by taking a probabilistic view. If we assume the risks are bounded in the third moment, w.l.o.g., the two risk-difference terms can both be generalized to a slightly more specific high-probability confidence interval [4, 9, 28] than the plain central limit theorem result itself: with probability of at least $1 - \alpha$, we have

$$||\tilde{R}_T(\{Q_1, ..., Q_T\}) - \tilde{R}_T(\{Q_1, ..., Q_T\} \setminus \mathcal{S}_{FB})||$$
$$\leq 2\left[F_{N_T}^{-1}\left(1 - \frac{\alpha}{2}\right)\frac{\tilde{\sigma}_{N_T}}{\sqrt{N_T}} - F_{N_T - N_{FB}}^{-1}\left(1 - \frac{\alpha}{2}\right)\frac{\tilde{\sigma}_{N_T - N_{FB}}}{\sqrt{N_T - N_{FB}}}\right] \tag{30}$$

$$||\tilde{R}_T(\{Q_1, ..., Q_T\}) - R|| \leq 2\left[F_{N_T}^{-1}\left(1 - \frac{\alpha}{2}\right)\frac{\tilde{\sigma}_{N_T}}{\sqrt{N_T}}\right] \tag{31}$$

where $F^{-1}$ is the inverse cumulative distribution function of the Student-$t$ distribution and $\tilde{\sigma}^2$ is the empirical variance. For the active feedback analysis, we only care about how $N_{FB}$ affects the testing objective, thus also obtaining an $\mathcal{O}(1/\sqrt{N_T - N_{FB}})$ dependency. $\qquad \square$

The detailed balancing between the two objectives (I) and (II) requires specific knowledge about the constants involved in the bounds. However, if we only focus on terms involving $N_{FB}$, both dependencies on the sample numbers are on the $1/\sqrt{n}$ level, making it possible to be balanced by a constant factor $C$. Combining these results, we get the $N_{FB}$ term as $\mathcal{O}(1/\sqrt{N_L + N_{FB}}) + \mathcal{O}(1/\sqrt{N_T - N_{FB}})$ (absorbing $\mathcal{O}(1)$ terms that do not depend on $N_{FB}$). The next key factor is that throughout the entire ATL process, we either keep $N_{FB}$ fixed or only change it at a linear rate (flexible $N_{FB}$ should be an interesting future direction). Combining with our previous assumption that $N_L$ and $N_T$ are of similar magnitudes, we know that an optimal balance could be achieved between (I) and (II) to minimize the joint learning-testing objective given in (8).

## C  Additional Experiment Results

In this section, we present the detailed experimental settings and additional experimental results.

### C.1  Synthetic Experiment

Figure 4 shows how the proposed feedback strategy helps to encourage exploration. The background color shows the model's predictive distribution. For each quiz, we display all the training samples obtained by an active learner (red and blue circles representing 2 classes) but only the current quiz (triangles) and feedback samples (squares, then added to circles in later AL rounds) from the active tester to make the visualization clear. Figure 4a shows that ATL selects a feedback sample in the bottom right corner because it is not included in the current knowledge base of the AL model. The AL model predicts it poorly in the quiz. In Figure 4b, we see that the AL model is guided by the feedback samples and starts to explore the bottom right corner. Once the AL model collects samples from the bottom right area, ATL stops to provide guidance for that region. In this way, the proposed feedback strategy manages to find the minority cluster at the other corner shortly as shown in Figure 4c.

In Figure 4d, to further demonstrate the effectiveness of the proposed feedback strategy, we compare it with the feedback samples selected using two other baselines: random feedback (in Figure 4f) and AL based feedback (in Figure 4e), when the samples at the bottom right corner are first discovered. First, we notice that those data samples are found by the AL model rather than through the feedback strategies. As a result, it happens at a much later quiz time compared with ATL. Therefore, they result in a less efficient learning process. Second, we observe that when an AL model discovers a

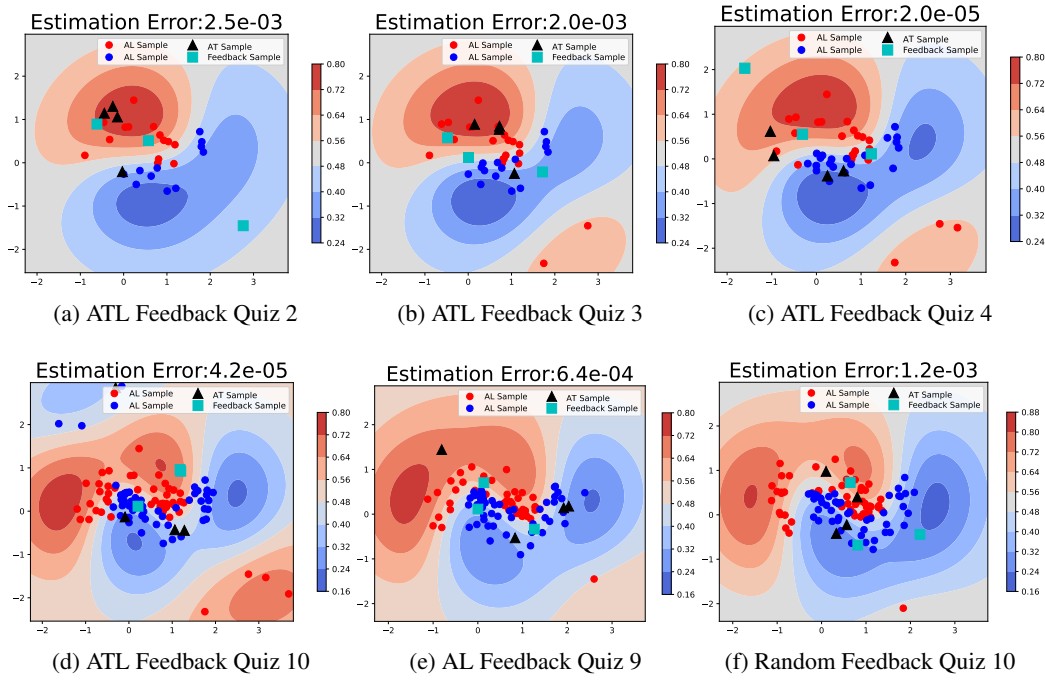

Figure 4: Effectiveness of active feedback for improving model training

Table 7: Test classification accuracy

| Feedback Strategy | Quiz 2 | Quiz 3 | Quiz 4 | Quiz 9 | Quiz 10 |
|---|---|---|---|---|---|
| ATL | 0.75 | 0.87 | 0.90 | 0.96 | 0.97 |
| AL | 0.75 | 0.83 | 0.84 | 0.84 | 0.88 |
| Random | 0.74 | 0.78 | 0.81 | 0.83 | 0.91 |
| ATL - NF | 0.74 | 0.76 | 0.79 | 0.81 | 0.86 |

new area to learn, both baseline feedback strategies fail to provide support even though they have some test samples (*i.e.,* the red point at the bottom right corner) available in the interesting region. Last, we can see that at around quiz 10, the AL model with the proposed ALT converges to a better decision boundary that captures the entire data distribution while the two other baselines both fail to correctly discover the predictive distribution at the two corners. As a result, ALT leads to a more accurate model (shown in Table 7) while maintaining lower estimation error in the end.

## C.2 Real-world Experiments

In this section, we provide more results on the real-world datasets including MNIST, FashionMNIST and CIFAR10, mainly to demonstrate different feedback approaches and how we can implement early stopping in ATL.

### C.2.1 Experimental Settings

In all experiments, we use a CNN model and standard data transformation for each dataset. In each AL training round, we run 10 epochs for MNIST and FashionMNIST and 50 epochs for CIFAR10. A threshold of $1 \times 10^{-5}$ is used for probability outputs as required for the proposal $q(\mathbf{x})$ computation [14] to avoid 0 denominators.

An important detail to note is that for **ATL-NF** results, we **sample 50 additional data points during AL for fair comparison** (550 in each round), which is actually very similar to **ATL-RF**. The results with only 500 data points per round will be shown in the following section C.2.2. Another detail worth mentioning is that although we set the initial budget to be $500$ labels and add $500$ training

samples plus 100 testing samples in each round, the final total budget is $12,450$ on average instead of $12,500$ because we allow replacement while sampling.

### C.2.2 Random Hold-out Test and Random Feedback

In this section, we discuss the issues with traditional hold-out validation/testing procedures during AL and compare the results using random sampling for both test sample selection and feedback selection with the proposed learning-testing-feedback process.

Table 8: Risk estimation error comparison with random methods

| Dataset | AL round / Method | 4 | 8 | 12 | 16 | 20 |
|---|---|---|---|---|---|---|
| MNIST | Random Test | $7.80 \pm 13.4$ | $6.61 \pm 3.72$ | $5.94 \pm 7.25$ | $3.15 \pm 5.83$ | $6.18 \pm 2.41$ |
| | Random Test & Feedback | $30.4 \pm 42.2$ | $16.8 \pm 16.1$ | $5.34 \pm 6.33$ | $11.8 \pm 10.6$ | $5.62 \pm 2.73$ |
| | Random Test & Weighted | $71.3 \pm 31.9$ | $19.1 \pm 16.5$ | $12.3 \pm 12.2$ | $10.0 \pm 11.7$ | $5.64 \pm 1.64$ |
| | ATL-NF | $\mathbf{2.57 \pm 1.17}$ | $\mathbf{0.79 \pm 1.15}$ | $\mathbf{0.17 \pm 0.15}$ | $\mathbf{0.56 \pm 0.30}$ | $\mathbf{1.32 \pm 0.37}$ |
| Fashion MNIST | Random Test | $6.97 \pm 11.2$ | $5.29 \pm 3.56$ | $10.9 \pm 6.55$ | $5.40 \pm 2.76$ | $8.56 \pm 7.57$ |
| | Random Test & Feedback | $12.7 \pm 13.4$ | $15.4 \pm 16.0$ | $12.7 \pm 6.93$ | $18.8 \pm 20.9$ | $32.7 \pm 15.4$ |
| | Random Test & Weighted | $6.85 \pm 14.8$ | $3.01 \pm 11.3$ | $2.80 \pm 11.1$ | $4.58 \pm 29.0$ | $9.51 \pm 9.13$ |
| | ATL-NF | $\mathbf{3.64 \pm 1.61}$ | $\mathbf{0.67 \pm 0.38}$ | $\mathbf{0.96 \pm 0.16}$ | $\mathbf{0.98 \pm 0.43}$ | $\mathbf{3.04 \pm 1.37}$ |
| CIFAR10 | Random Test | $20.5 \pm 6.50$ | $15.8 \pm 10.3$ | $13.0 \pm 10.1$ | $9.99 \pm 6.58$ | $9.89 \pm 9.61$ |
| | Random Test & Feedback | $44.7 \pm 36.4$ | $16.4 \pm 15.4$ | $31.0 \pm 12.8$ | $11.7 \pm 8.65$ | $55.3 \pm 19.0$ |
| | Random Test & Weighted | $43.5 \pm 14.8$ | $14.6 \pm 11.3$ | $15.1 \pm 11.1$ | $11.2 \pm 29.0$ | $48.7 \pm 9.13$ |
| | ATL-NF | $\mathbf{8.83 \pm 7.79}$ | $\mathbf{3.06 \pm 5.04}$ | $\mathbf{4.95 \pm 7.12}$ | $\mathbf{7.94 \pm 5.22}$ | $\mathbf{6.20 \pm 5.79}$ |
| Synthetic | Random Test | $4.17 \pm 0.78$ | $8.00 \pm 0.71$ | $11.2 \pm 0.19$ | $5.06 \pm 0.10$ | $8.96 \pm 0.04$ |
| | Random Test & Feedback | $67.8 \pm 2.7$ | $75.0 \pm 3.95$ | $35.6 \pm 1.02$ | $8.29 \pm 1.29$ | $0.51 \pm 0.72$ |
| | Random Test & Weighted | $5.26 \pm 2.04$ | $14.10 \pm 2.51$ | $10.66 \pm 1.85$ | $8.71 \pm 1.02$ | $9.21 \pm 0.07$ |
| | ATL-NF | $\mathbf{3.67 \pm 0.11}$ | $\mathbf{2.86 \pm 0.63}$ | $\mathbf{1.48 \pm 0.14}$ | $\mathbf{1.20 \pm 0.02}$ | $\mathbf{0.26 \pm 0.07}$ |

In Table 8, random test is referring to randomly sampling 100 test samples after each 550-sample (additional 50 for fair comparison, same as ATL-NF) AL round and simply averaging the loss over these test samples. Random test & feedback is referring to sampling 100 test samples after each 500-sample AL round and then randomly selecting 50 for feedback. Random test & weighted is referring to the same process but the quizzes are weighted by $1/R_t$. From Table 8, we can see that in the small-data regime, random sampling may not provide an accurate estimate of the true risk. However, in later AL rounds, the no feedback case (Random Test) can maintain an unbiased estimate, and we do see that some results are comparable with active risk estimation baselines without the ATL-integrate estimator. This is probably because existing active risk estimation baselines (ARE-quiz, AT-integrate, ASE-integrate) do not consider the biased selection and model change through the AL process. The methods that use surrogate models also suffers from the insufficient training of the surrogate model. However, random testing selection does not work well with the active feedback process. For Random Test & Feedback and Random Test & Weighted, we often see much worse estimation due to the feedback process involved.

### C.2.3 ATL with Various AL Strategies

In this section, we show additional ATL estimation error results and active feedback results using two different AL strategies: margin sampling [27] and BALD [12].

From Table 9, we can see that the integrated risk estimation performs similarly regardless of the AL strategy. The relative scales of different risk estimations are also similar to the entropy case in the main paper. From Table 10, we do see some different effects. Depending on the AL strategy, random feedback performs unstably. However, our proposed feedback still mostly outperforms the fair comparison baseline. The study on AL-specific feedback strategies could be an interesting future direction.

### C.2.4 Additional Active Feedback Comparisons

In this section, we show a more complete comparison between different feedback approaches. The feedback comparison consists of two parts: (1) baseline comparison including no feedback (ATL-NF), random feedback (ATL-RF), entropy-based feedback (ATL-EN) and (2) ablation study including loss-based feedback (ATL-LF), weighted loss-based feedback (ATL-WL) and the proposed weighted loss plus diversity feedback (ATL).

Table 9: Squared difference between the estimate and the true risk over 20 AL rounds ($\times 10^{-3}$)

| Dataset | AL strategy | AL round / Method | 4 | 8 | 12 | 16 | 20 |
|---|---|---|---|---|---|---|---|
| MNIST | Margin | ARE quiz | 16.9 | 2.30 | 3.86 | 6.91 | 10.1 |
| | | AT integrate | 11.7 | 18.9 | 15.3 | 8.35 | 2.64 |
| | | ASE integrate | 3.96 | 3.78 | 5.04 | 11.1 | 17.6 |
| | | ATL | **3.40** | **3.49** | **3.94** | **1.93** | **2.46** |
| Fashion MNIST | Margin | ARE quiz | 16.3 | 16.7 | 11.5 | 11.8 | 9.92 |
| | | AT integrate | 6.29 | 5.66 | 6.98 | 7.04 | 2.55 |
| | | ASE integrate | 18.4 | 12.5 | 19.9 | 13.4 | 5.38 |
| | | ATL | **1.14** | **0.95** | **0.87** | **0.79** | **0.62** |
| CIFAR10 | Margin | ARE quiz | 11.1 | 3.31 | 1.32 | 1.24 | 5.64 |
| | | AT integrate | 3.97 | 7.05 | 7.48 | 6.74 | 8.33 |
| | | ASE integrate | 9.15 | 12.8 | 3.99 | 9.50 | 4.04 |
| | | ATL | **2.49** | **0.87** | **1.88** | **0.55** | **2.96** |
| MNIST | BALD | ARE quiz | 16.99 | 9.16 | 5.75 | 6.91 | 10.1 |
| | | AT integrate | 8.48 | 35.9 | 11.7 | 28.5 | 19.1 |
| | | ASE integrate | 16.4 | 5.52 | 7.88 | 2.16 | 4.81 |
| | | ATL | **3.83** | **4.41** | **5.24** | **1.54** | **2.82** |
| Fashion MNIST | BALD | ARE quiz | 11.1 | 12.2 | 10.1 | 13.1 | 9.92 |
| | | AT integrate | 6.09 | 4.04 | 7.33 | 8.86 | 8.75 |
| | | ASE integrate | 9.00 | 10.7 | 4.73 | 5.23 | 5.69 |
| | | ATL | **4.25** | **3.92** | **4.50** | **4.28** | **4.59** |
| CIFAR10 | BALD | ARE quiz | 3.04 | 8.93 | 3.11 | 19.4 | 13.2 |
| | | AT integrate | 12.5 | 22.3 | 21.9 | 15.6 | 14.9 |
| | | ASE integrate | 2.47 | 3.94 | 5.20 | 12.0 | 19.7 |
| | | ATL | **1.09** | **3.82** | **4.65** | **7.21** | **10.6** |

Table 10: Hold-out test risk using different AL strategies and feedback methods over 20 AL rounds

| Dataset | AL strategy | AL round / Method | 4 | 8 | 12 | 16 | 20 |
|---|---|---|---|---|---|---|---|
| MNIST | Margin | ATL-NF | 0.97 | 0.61 | 0.49 | 0.29 | 0.14 |
| | | ATL-RF | 0.89 | 0.52 | 0.32 | 0.22 | 0.14 |
| | | ATL | **0.88** | **0.53** | **0.35** | **0.17** | **0.14** |
| Fashion MNIST | Margin | ATL-NF | 1.30 | 1.05 | 0.90 | 0.82 | 0.69 |
| | | ATL-RF | 1.23 | 1.06 | 0.88 | 0.73 | 0.70 |
| | | ATL | **1.14** | **0.95** | **0.87** | **0.79** | **0.62** |
| CIFAR10 | Margin | ATL-NF | 1.68 | 1.57 | 1.50 | 1.55 | 1.37 |
| | | ATL-RF | 1.61 | 1.58 | 1.56 | 1.57 | 1.34 |
| | | ATL | **1.62** | **1.55** | **1.49** | **1.54** | **1.34** |
| MNIST | BALD | ATL-NF | 1.03 | 0.75 | 0.49 | 0.36 | 0.29 |
| | | ATL-RF | 0.91 | 0.58 | 0.44 | 0.35 | 0.25 |
| | | ATL | **0.89** | **0.59** | **0.46** | **0.35** | **0.25** |
| Fashion MNIST | BALD | ATL-NF | 1.18 | 1.04 | 0.91 | 0.89 | 0.85 |
| | | ATL-RF | 1.40 | 1.06 | 0.95 | 0.88 | 0.81 |
| | | ATL | **1.19** | **1.07** | **0.90** | **0.83** | **0.80** |
| CIFAR10 | BALD | ATL-NF | 1.95 | 1.92 | 1.78 | 1.72 | 1.72 |
| | | ATL-RF | 1.93 | 1.91 | 1.79 | 1.75 | 1.73 |
| | | ATL | **1.88** | **1.80** | **1.76** | **1.56** | **1.57** |

First, we show the hold-out test risk of the AL model throughout AL using different active feedback approaches as the indicator of the model performance. From Table 11, we see that in most occasions, all active feedback approaches can reduce the test risk compared to ATL-NF.

In Table 12, we show a full comparison of the squared error of risk estimation. All estimation results are based on the proposed ATL estimator $\tilde{R}$, where ATL-NF, ATL-RF, ATL-EN serve as baselines, meanwhile ATL-LF and ATL-WL serve as ablation studies since the proposed ATL utilizes the weighted loss as well. We see that all feedback approaches suffer from an increased estimation error, especially in the early stage when the number of test samples available is small. We see that

Table 11: Hold-out test risk using different feedback criteria over 20 AL rounds

| Dataset | Method \ AL round | 4 | 8 | 12 | 16 | 20 |
|---|---|---|---|---|---|---|
| MNIST | ATL-NF | $0.92 \pm 0.06$ | $0.55 \pm 0.08$ | $0.46 \pm 0.06$ | $0.32 \pm 0.04$ | $0.22 \pm 0.02$ |
| | ATL-RF | $0.92 \pm 0.12$ | $0.54 \pm 0.02$ | $0.41 \pm 0.05$ | $0.29 \pm 0.03$ | $0.21 \pm 0.02$ |
| | ATL-EN | $0.90 \pm 0.12$ | $0.55 \pm 0.06$ | $0.41 \pm 0.02$ | $0.34 \pm 0.06$ | $0.23 \pm 0.03$ |
| | ATL-LF | $0.89 \pm 0.10$ | $0.56 \pm 0.04$ | $0.41 \pm 0.02$ | $0.32 \pm 0.07$ | $0.20 \pm 0.02$ |
| | ATL-WL | $0.86 \pm 0.06$ | $0.53 \pm 0.06$ | $0.40 \pm 0.05$ | $0.32 \pm 0.07$ | $0.22 \pm 0.03$ |
| | ATL | $\mathbf{0.88 \pm 0.07}$ | $\mathbf{0.53 \pm 0.04}$ | $\mathbf{0.39 \pm 0.03}$ | $\mathbf{0.26 \pm 0.01}$ | $\mathbf{0.19 \pm 0.03}$ |
| Fashion MNIST | ATL-NF | $0.75 \pm 0.03$ | $0.69 \pm 0.02$ | $0.61 \pm 0.02$ | $0.57 \pm 0.04$ | $0.56 \pm 0.03$ |
| | ATL-RF | $0.75 \pm 0.04$ | $0.68 \pm 0.02$ | $0.61 \pm 0.01$ | $0.58 \pm 0.06$ | $0.56 \pm 0.04$ |
| | ATL-EN | $0.76 \pm 0.02$ | $0.67 \pm 0.05$ | $0.58 \pm 0.02$ | $0.59 \pm 0.03$ | $0.56 \pm 0.02$ |
| | ATL-LF | $0.76 \pm 0.04$ | $0.65 \pm 0.03$ | $0.63 \pm 0.01$ | $0.56 \pm 0.02$ | $0.56 \pm 0.04$ |
| | ATL-WL | $0.76 \pm 0.03$ | $0.65 \pm 0.02$ | $0.62 \pm 0.01$ | $0.56 \pm 0.02$ | $0.53 \pm 0.02$ |
| | ATL | $\mathbf{0.74 \pm 0.03}$ | $\mathbf{0.65 \pm 0.04}$ | $\mathbf{0.59 \pm 0.02}$ | $\mathbf{0.56 \pm 0.03}$ | $\mathbf{0.51 \pm 0.01}$ |
| CIFAR10 | ATL-NF | $1.91 \pm 0.04$ | $1.76 \pm 0.05$ | $1.72 \pm 0.01$ | $1.66 \pm 0.02$ | $1.55 \pm 0.03$ |
| | ATL-RF | $1.91 \pm 0.03$ | $1.77 \pm 0.04$ | $1.69 \pm 0.03$ | $1.60 \pm 0.04$ | $1.54 \pm 0.07$ |
| | ATL-EN | $1.92 \pm 0.09$ | $1.76 \pm 0.04$ | $1.70 \pm 0.03$ | $1.66 \pm 0.04$ | $1.54 \pm 0.02$ |
| | ATL-LF | $1.94 \pm 0.04$ | $1.75 \pm 0.03$ | $1.65 \pm 0.01$ | $1.59 \pm 0.03$ | $1.54 \pm 0.01$ |
| | ATL-WL | $1.94 \pm 0.04$ | $1.75 \pm 0.03$ | $1.63 \pm 0.01$ | $1.63 \pm 0.03$ | $1.54 \pm 0.01$ |
| | ATL | $\mathbf{1.90 \pm 0.05}$ | $\mathbf{1.76 \pm 0.02}$ | $\mathbf{1.65 \pm 0.03}$ | $\mathbf{1.58 \pm 0.02}$ | $\mathbf{1.53 \pm 0.02}$ |

Table 12: Squared difference between the estimate and the true risk over 20 AL rounds ($\times 10^{-3}$)

| Dataset | Method \ AL round | 4 | 8 | 12 | 16 | 20 |
|---|---|---|---|---|---|---|
| MNIST | ATL-NF | $2.57 \pm 1.17$ | $0.79 \pm 1.15$ | $0.17 \pm 0.15$ | $0.56 \pm 0.30$ | $1.32 \pm 0.37$ |
| | ATL-RF | $26.8 \pm 21.4$ | $21.4 \pm 17.0$ | $3.54 \pm 4.01$ | $5.54 \pm 3.21$ | $7.62 \pm 4.41$ |
| | ATL-EN | $23.6 \pm 24.8$ | $14.0 \pm 15.8$ | $13.8 \pm 11.7$ | $29.5 \pm 21.7$ | $21.8 \pm 12.8$ |
| | ATL-LF | $15.6 \pm 12.6$ | $42.4 \pm 36.9$ | $48.5 \pm 25.8$ | $15.7 \pm 14.8$ | $10.9 \pm 7.44$ |
| | ATL-WL | $16.5 \pm 19.4$ | $21.0 \pm 24.3$ | $7.36 \pm 8.44$ | $11.4 \pm 12.7$ | $7.59 \pm 4.45$ |
| | ATL | $\mathbf{14.6 \pm 22.1}$ | $\mathbf{16.9 \pm 13.7}$ | $\mathbf{3.19 \pm 2.63}$ | $\mathbf{4.15 \pm 3.20}$ | $\mathbf{1.87 \pm 1.41}$ |
| Fashion MNIST | ATL-NF | $3.64 \pm 1.61$ | $0.67 \pm 0.38$ | $0.96 \pm 0.16$ | $0.98 \pm 0.43$ | $3.04 \pm 1.37$ |
| | ATL-RF | $10.2 \pm 9.30$ | $4.41 \pm 3.77$ | $2.19 \pm 5.53$ | $5.69 \pm 4.52$ | $11.6 \pm 7.51$ |
| | ATL-EN | $93.2 \pm 23.4$ | $50.2 \pm 10.2$ | $78.5 \pm 32.4$ | $76.2 \pm 59.6$ | $85.8 \pm 25.9$ |
| | ATL-LF | $9.36 \pm 10.2$ | $27.2 \pm 26.0$ | $22.6 \pm 28.3$ | $14.6 \pm 12.0$ | $11.0 \pm 15.2$ |
| | ATL-WL | $8.39 \pm 8.97$ | $7.52 \pm 6.09$ | $4.89 \pm 6.50$ | $7.29 \pm 4.45$ | $11.1 \pm 7.02$ |
| | ATL | $\mathbf{2.50 \pm 2.93}$ | $\mathbf{1.94 \pm 2.25}$ | $\mathbf{1.78 \pm 1.07}$ | $\mathbf{6.32 \pm 5.41}$ | $\mathbf{5.03 \pm 4.41}$ |
| CIFAR10 | ATL-NF | $8.83 \pm 7.79$ | $3.06 \pm 5.04$ | $4.95 \pm 7.12$ | $7.94 \pm 5.22$ | $6.20 \pm 5.79$ |
| | ATL-RF | $20.6 \pm 17.6$ | $19.1 \pm 13.7$ | $9.82 \pm 8.03$ | $33.6 \pm 30.5$ | $24.8 \pm 32.4$ |
| | ATL-EN | $30.3 \pm 17.0$ | $45.8 \pm 24.4$ | $20.3 \pm 17.4$ | $36.8 \pm 31.7$ | $27.0 \pm 27.1$ |
| | ATL-LF | $35.0 \pm 27.9$ | $45.8 \pm 28.5$ | $20.3 \pm 10.1$ | $57.2 \pm 33.6$ | $40.5 \pm 34.0$ |
| | ATL-WL | $22.7 \pm 19.7$ | $25.0 \pm 13.2$ | $12.9 \pm 21.5$ | $52.2 \pm 45.9$ | $28.7 \pm 16.3$ |
| | ATL | $\mathbf{11.6 \pm 13.4}$ | $\mathbf{5.11 \pm 3.45}$ | $\mathbf{8.81 \pm 6.51}$ | $\mathbf{11.9 \pm 16.7}$ | $\mathbf{6.57 \pm 6.29}$ |

the baseline methods suffer from increased estimation error. However, ATL can usually maintain a similar level of estimation error after 20 AL rounds. For ATL-LF, there is usually a larger variance of the estimation error. The potential reason for the unstable behavior of ATL-LF is that it only selects samples with larger losses in the feedback process. Although the importance mechanism can make up for some of the difference, there is still the potential risk of the estimate being biased. Further combining with the diversity metric, we achieve the best results with ATL.

Concluding from both the risk results and the estimation error results, we show that the proposed feedback approach achieves a good balance in the performance-estimation trade-off. This is because we consider both the loss $L$ and the importance weight $q$ in the selection criterion. Overall, ATL achieves a similar model test risk as ATL-LF/ATL-WL, both of which are much better than ATL-NF and ATL-RF. ATL also achieves a much lower estimation error than ATL-RF, ATL-EN, and ATL-LF.

### C.2.5 Single Feedback Round Comparison

In previous experiments, we add additional training points for the no feedback case (ATL-NF) to make fair comparison for the model risk. However, if we look at the risk change before and after a single feedback round, the difference is even more obvious as shown in Table 13.

### C.2.6 Early Stopping in AL

In this section, we show how the ATL-based risk estimation can be readily used for early stopping in AL. In the above experiments, we observe a steady decrease of the estimated risk most of the

Table 13: Hold-out test risk before and after a specific feedback round

| Dataset | AL round / Method | 4 | 8 | 12 | 16 | 20 |
|---|---|---|---|---|---|---|
| MNIST | ATL-before | $0.91 \pm 0.09$ | $0.54 \pm 0.04$ | $0.41 \pm 0.08$ | $0.29 \pm 0.02$ | $0.21 \pm 0.03$ |
| | ATL-after | $\mathbf{0.88 \pm 0.07}$ | $\mathbf{0.53 \pm 0.04}$ | $\mathbf{0.39 \pm 0.03}$ | $\mathbf{0.26 \pm 0.01}$ | $\mathbf{0.19 \pm 0.03}$ |
| Fashion MNIST | ATL-before | $0.77 \pm 0.03$ | $0.66 \pm 0.03$ | $0.61 \pm 0.02$ | $0.57 \pm 0.03$ | $0.53 \pm 0.03$ |
| | ATL-after | $\mathbf{0.74 \pm 0.03}$ | $\mathbf{0.65 \pm 0.04}$ | $\mathbf{0.59 \pm 0.02}$ | $\mathbf{0.56 \pm 0.03}$ | $\mathbf{0.51 \pm 0.01}$ |
| CIFAR10 | ATL-before | $1.97 \pm 0.07$ | $1.82 \pm 0.05$ | $1.70 \pm 0.03$ | $1.67 \pm 0.03$ | $1.57 \pm 0.04$ |
| | ATL-after | $\mathbf{1.90 \pm 0.05}$ | $\mathbf{1.76 \pm 0.02}$ | $\mathbf{1.65 \pm 0.03}$ | $\mathbf{1.58 \pm 0.02}$ | $\mathbf{1.53 \pm 0.02}$ |

times. However, we do find the decrease becomes more insignificant near the end of the 20 rounds of learning, especially for the MNIST and Fashion MNIST datasets. We observe that after a certain amount of AL rounds, the risk decrease is significantly small, and the corresponding test accuracy is also stabilized (MNIST around $94\%$, Fashion MNIST around $80\%$, CIFAR around $54\%$). This gives us the opportunity to apply early stopping in real-life AL applications. We here show the average stopping iteration and model performance (hold-out test accuracy) of the compared methods in Table 14. Following the same threshold value, by augmenting the moving average of active risk estimation given by (10) with stabilized prediction (SP), the combined method can stop at a similar testing accuracy as compared with the SP method, but with much lower variance in test accuracy. Based on the threshold setting, it is also possible to stop AL much earlier, saving the overall labeling budget.

Table 14: Average early stopping iteration and final test accuracy comparison (with variance)

| Dataset | Method | Iteration | Variance | Test Accuracy | Variance |
|---|---|---|---|---|---|
| MNIST | SP | 15 | 6.8 | $94.52\%$ | $6.0e-5$ |
| | Combined | 11 | 1.2 | $94.08\%$ | $3.1e-5$ |
| Fashion MNIST | SP | 16 | 4.4 | $81.32\%$ | $3.7e-5$ |
| | Combined | 12.4 | 1.04 | $80.12\%$ | $2.4e-5$ |
| CIFAR10 | SP | 12 | 2.8 | $53.87\%$ | $1.4e-4$ |
| | Combined | 12.8 | 0.16 | $54.43\%$ | $8.9e-5$ |

## D  Details of Hardware for Experiments

All experiments were run on clusters with either NVIDIA A6000 or NVIDIA A100 graphic cards and Intel Xeon Gold 6150 CPU processors. The runtime of the experiments varies depending on the number of repeat runs, but is usually on the scale of a few hours. For example, to get the 5 runs results of one ATL setting for 20 AL rounds on MNIST or Fashion MNIST may take about 6 to 8 hours. The CIFAR10 experiments may take slightly longer.

## E  Limitation, Future work, and Social Impact

In this section, we first discuss some limitation of the proposed framework and identify some important future direction. We then discuss some potential social impact of our work.

### E.1  Limitation and Future Directions

In this paper, we propose an integrated framework that combines active learning and testing. In the interactive framework, the exchange of training and testing information should be carefully guided. Although the proposed testing selection is statistically unbiased and the active feedback is backed by the high-level analysis, we still have room for improving the specification of methods in applicable settings, which we will introduce here as future directions:

- From the learning perspective, we can improve upon the general setting in this paper. In this paper, we focus on introducing a general framework and working under the agnostic setting. However, using specific AL strategies can potentially provide advantages in certain use cases. There have been works that analyze AL label complexity bounds using either importance weighting mechanism in stream-based settings [5, 8] or other methods in pool-based settings [11].

- Continuing on the results from the feedback size analysis in Section 4.3 and the discussion above, the feedback size is a very important factor in the process, especially if we allow the

size to change during AL. Further investigating the relationship between the sample size and the combined learning-testing objective can potentially improve the framework.

- We also propose the ATL framework under an AL-agnostic assumption. Given specific AL strategies, we might be able to also incorporate the learning or testing proposal in the construction of feedback proposal.

### E.2 Social Impact

The proposed ATL framework considers the practical challenges of applying active learning in real-world settings, where both model training and evaluation require labeled data. It is a critical step towards realizing label-efficient learning in practice, which can benefit many critical domains where data annotation is highly costly. To this end, the proposed ATL framework has the potential to fundamentally address the data annotation crisis and further broaden the usage of AI to benefit the entire society.

## F Algorithm and Source Code

Here we present the pseudo code for the ATL framework using an algorithm block below:

---

**Algorithm 1:** Active Testing While Learning (ATL)

---

**Input** : Total number of quizzes: $T$,
Active learning/testing/feedback budget: $N_{AL}, N_{AT}, N_{FB}$
Unlabeled pool: $S_U$,
Active learning model at quiz $t$: $f_{\theta_t}(\mathbf{x})$,
AL sampling strategy: $a : f_{\theta_t}(\mathbf{x}) \times S_U \to \mathbb{R}$,
Learning objective $\mathcal{L}$: $\mathcal{L} : f_{\theta_t}(\mathbf{x}) \times y \to \mathbb{R}$,
Annotation method: $h : \mathbf{x} \to y$
**Output** : Quiz samples $\mathcal{Q} = \{\mathcal{Q}_1, \mathcal{Q}_2, ..., \mathcal{Q}_T\}$,
Quiz results: $R_Q = \{R_{\mathcal{Q}_1}, R_{\mathcal{Q}_2}, ..., R_{\mathcal{Q}_T}\}$,
Annotated training dataset: $S_L$

1   $S_L = \{\}$           `// Active Learning Samples`
2   $\mathcal{Q} = \{\}$           `// Quiz Selection Criterion`
3   **for** $t = 1$ **to** $T$ **do**
4      **for** $i = 1$ **to** $N_{AL}$ **do**
5          $\mathbf{x}_i = \underset{\mathbf{x}_i}{\operatorname{argmax}}\, a(f_{\theta_t}, S_U)$      `// Active Learning Starts`
6          $S_L = S_L \cup \{\mathbf{x}_i, h(\mathbf{x}_i)\}$
7          $S_U = S_U \backslash \{\mathbf{x}_i\}$      `// Active Learning Ends`
8      Estimate $R_\theta^{\mathrm{multi}}$ using eq(5)      `// Active Testing Starts`
9      Compute test distribution $q()$ using eq(4)
10     $\mathcal{Q}_t = \{\}$
11     **for** $i = 1$ **to** $N_{AT}$ **do**
12        Select test sample $\mathbf{x}_i \sim q(\mathbf{x}_i, R_\theta^{\mathrm{multi}}, S_U)$
13        $\mathcal{Q}_t = \mathcal{Q}_t \cup \{\mathbf{x}_i\}$
14     $S_U = S_U \backslash \mathcal{Q}_t$
15     **for** $i = 1$ **to** $N_{FB}$ **do**
16        Choose $\mathbf{x}^*$ using eq(9)      `// Active Feedback Starts`
17        $S_L = S_L \cup \{\mathbf{x}^*\}$
18        $\mathcal{Q}_t = \mathcal{Q}_t \backslash \{\mathbf{x}^*\}$      `// Active Feedback Ends`
19     Compute $\widehat{R}_{\mathcal{Q}_t}$ using eq(6)
20     $R_Q = R_Q \cup \{\widehat{R}_{\mathcal{Q}_t}\}$
21     $\mathcal{Q} = \mathcal{Q} \cup \mathcal{Q}_t$      `// Active Testing Ends`

---

**Source code**. The data and source code for replicating the results are provided in this link: `https://github.com/ritmininglab/ATL.git`

