# Appendix

## Table of Contents

## A Appendix Overview

**Organization.** The Appendix is organized as follows. We first provide a summary of notations in Appendix B. Then, we present the details of the theoretical analysis in the main paper in Appendix C. Next, we show additional experimental results in Appendix D, including a synthetic visualization for different feedback approaches in Appendix D.1, the risk and estimation error comparison on real-world datasets in Appendix D.2.3 and the early stopping experiments

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

usp=sharing
```