# OpenReview forum: "Actively Testing Your Model While It Learns: Realizing Label-Efficient Learning in Practice"
_NeurIPS.cc/2023/Conference — NeurIPS 2023 poster_

### Official Review · Reviewer_2XK2 · 2023-06-30

**Soundness:** 2 fair
**Presentation:** 4 excellent
**Contribution:** 4 excellent
**Rating:** 6
**Confidence:** 3

**Summary:**

The paper introduces the Active Testing while Learning (ATL) framework, which efficiently collects testing samples for online model risk estimation during the active learning process. The online risk estimation enables early termination of active learning. Additionally, ATL establishes a connection between Active Learning (AL) and Active Testing (AT) through an active feedback mechanism that transfers selected testing data points to the training dataset, thereby enhancing model performance.

**Strengths:**

1. The paper tackles two crucial problems in the AL domain: efficient sampling of testing data points and the combination of AL with AT. Both of these problems are highly relevant and significant in AL research.
2. The paper introduces an unbiased estimator of the model risk, and also proposes a unique method to combine active quizzes as the final exam, maximizing the utilization of testing samples.
3. The paper provides a comprehensive set of experiments to evaluate the proposed estimator and feedback methods. The experimental results demonstrate the superior performance of the proposed approaches compared to existing techniques.

**Weaknesses:**

In the first half, the paper focuses extensively on optimizing the data sampling process to obtain an unbiased estimator of model risk. However, a flaw arises in the feedback mechanism, which moves data points from testing dataset to the training dataset based on the rule described in equations (8) or (9). This feedback undermines the unbiasedness of the risk estimator by introducing dependence in the selection of testing samples. Specifically, when we extract samples from the testing datasets based on some specific rule such as (8) or (9), the remaining samples are no longer independently sampled, thereby introducing bias.

**Questions:**

The paper needs more discussion on the potential bias introduced by the feedback mechanism.

---

> ### Author Rebuttal · Authors · 2023-08-10
>
> **Q1: Bias introduced by the feedback mechanism.**
>
> Thank you very much for raising this insightful point! As we agree with your comment, we here further elaborate on the problem following the answer in Q5.
>
> The key objective of feedback is to improve model learning without sacrificing the risk estimation too much. In Section 3.5, we investigate active feedback as a way to optimize a (novel) joint objective of learning and testing. Our theoretical result reveals that it is possible to achieve an optimal balance between the two sub-goals (i.e., (I) and (II) as specified in Eq (8) of the paper) by choosing a suitable feedback set such that we can further improve the model learning performance while maintaining the risk estimation quality from our quizzes-testing process. Meanwhile, given the AL-agnostic nature of the framework, no further assumptions on the actual AL algorithms are available, thus leaving a general analysis.
>
> Following the general analysis of the combined objective of learning and testing, we focus on proposing a practical solution. To minimize the negative impact of the feedback process on the risk estimation, we propose a practical solution in Eq.(9) that considers the corresponding test sampling importance $q({\bf x})$ of test samples. It aims to improve the combined objective of learning and testing in active feedback. The empirical results in Table 3 also show that we maintain the risk estimation better than random feedback sampling.
>
> In the revised paper, we will further clarify the limitation of the feedback process and clearly state the further analysis of the feedback impact as part of future works.

---

> ### Author Response · Authors · 2023-08-18
>
> Thank you again for your review and suggestions. We appreciate the comments about the importance of the unbiasedness analysis. We have incorporated your suggestions and further clarified the contribution including the unbiased risk estimator for integrated quizzes over AL and the high-level analysis plus practical solution for active feedback. We sincerely hope that you can find our answers satisfactory and consider updating the assessment. We are happy to provide any additional information if needed.

---

### Official Review · Reviewer_5t4X · 2023-07-07

**Soundness:** 3 good
**Presentation:** 3 good
**Contribution:** 2 fair
**Rating:** 5
**Confidence:** 4

**Summary:**

This paper considers the active testing procedure to assess the risk. The keys are 1) a proposal of a quiz using importance sampling, 2) combining the historical quizzes to make a test set for the final model test, and 3) a proposal of a feedback algorithm to strengthen the model using the labeled test data. Experiments focus on the performance of risk estimation, and some work concern test accuracy with feedback algorithms.

**Strengths:**

The proposal of an activel testing procedure can catch realistic problems. The quiz usage is impressive, and the set of quizzes is fully used.

**Weaknesses:**

Maybe we can consider a more straightforward example, such as the explicitly calculated risk. The performance of the proposed algorithms is validated. However, more solid experiment results are limited since the active learning algorithms are various, but only a few algorithms are considered.

**Questions:**

I have some questions.

1) what's the exact AL algorithm used in the experiments?

2) Is there any effect of different AL algorithms on ALT?

3) the feedback algorithm seems to be a hybrid of AT and ATL. Therefore, it seems that the benefit of feedback cannot be larger than the pure AL in the same budget (not the same budget in experiments)?

4) Is the algorithm possibly adaptive to the quiz sets (test sets)? I want to know how to construct the test set in Table 2.

**Limitations:**

Moderate explained. The connection between AL and AT seems lower.

---

> ### Author Rebuttal · Authors · 2023-08-10
>
> **Q1: Explicitly calculated risk and limitation from AL algorithm.**
>
> We would like to clarify that our design of framework is AL-agnostic, and we do not assume any fixed distribution for AL sampling. We will further discuss the AL strategies in Q2 and Q3 below. As for the test risk specification, in the evaluation, we do have the hold-out test risk as the reference true risk (we will further explain in Q5 below).
>
> **Q2\&Q3: AL strategy specification and various strategies.**
>
> Please see Q2 in the general response. Since our integrated risk estimator is AL-agnostic, different AL strategies do not have an effect on the risk estimation. However, the choice of AL strategies might affect how much the active feedback approach can improve the training. As we see from the results, the relative comparisons among different risk estimations are similar when using various AL strategies. Although random feedback might not be as effective when using BALD,  our proposed feedback still mostly outperforms the fair comparison baseline.
>
> **Q4: The benefit of feedback cannot be larger than pure AL.**
>
> We would like to clarify that in our framework, both learning and validation are considered to be sample-efficient and we consider the total labeling budget. As we are not certain about the "pure AL in the same budget" setting description, to respond to the comment, we would like to discuss two different cases: 1) pure AL with full budget vs ATL; 2) fair comparison between AL and ATL. (For confusion about the hold-out test set used for evaluation, please see Q5 below.)
>
> 1) In the first case, we assume that pure AL uses the entire budget $N_B$ for training, while ATL needs to divide the budget into $N_L$ and $N_T$ ($N_{FB}$ will be taken from $N_T$ and added to $N_L$). Indeed, the benefit of feedback can not be larger than using all labeling budget for AL. However, we are assuming that some validation is needed during the AL process, and there is no way to perform validation if the labeling budget is drained by AL. Thus, our goal is a combined objective of improving the training performance while not sacrificing too much of the estimation quality.
>
> 2) In the second case, we compare AL using the fair budget $(N_{L})^*=N_{L}+N_{FB}$. This is the setting we use in Table 2 and Appendix D.2.3-D.2.4, Tables 7 and 8. In this case, whenever we compare the model performance (hold-out test risk), the model is trained using the same number of labels for ATL-NF and all feedback methods. In this case, we do not think a clear conclusion can be drawn about whether the benefit of feedback can be larger than pure AL. On the one hand, we have a limited selection range. On the other hand, we have already obtained labels for all samples within $N_T$, thus could potentially choose better samples. We acknowledge that this could be tightly related to the specific AL strategy, but at least in our experiments using entropy sampling, feedback results can in fact surpass pure AL results in most cases (Table 7). We do provide an additional study on the various percentage of labeling budget being used for active feedback in Tables 9 and 10 in Appendix D.2.4, which shows that the relationship is not trivial. We consider the further study on the trade-off between learning performance and risk estimation quality as an important future direction.
>
> **Q5: Constructing test sets and the evaluation setting.**
>
> For the first part of the question, we do make an adaptation of active risk estimation approaches to the AL setting, where we formulate the integrated risk estimator from individual quizzes, aiming to minimize the estimation variance. For the second part of the question, the hold-out test set is fixed using a large number of samples to provide the reference true risk. The hold-out test risk is solely used for evaluation purposes. (Similar to AL works that still use the hold-out test accuracy as the evaluation method, we also need to have a hold-out test set to provide the reference true risk. The difference is that in pure AL, the hold-out test set only provides reference for the risk/accuracy performance, while here it is also used to compute the risk estimation error.) This should come naturally to any AL/AT work.

---

> ### Author Response · Authors · 2023-08-18
>
> Thank you again for your review and suggestions. In particular, we have further clarified the problem setting and the ATL framework. We appreciate the question about the benefit of active feedback and hope we have addressed it well. As we have also provided the missing results and necessary information in the rebuttal, we sincerely hope that you can find our answers satisfactory and consider updating the assessment. We are happy to provide any additional information if needed.

---

> > ### Comment · Reviewer_5t4X · 2023-08-21
> > **Response**
> >
> > I carefully read your responses. All concerned issues are well addressed, and I have no further questions. I'll raise my score.

---

> > > ### Author Response · Authors · 2023-08-21
> > >
> > > Thank you very much for your positive feedback. We are happy to know that our responses have addressed your concerns. Your insights are instrumental in shaping the final version of our work.

---

### Official Review · Reviewer_bQCr · 2023-07-07

**Soundness:** 3 good
**Presentation:** 3 good
**Contribution:** 3 good
**Rating:** 6
**Confidence:** 3

**Summary:**

The paper provides a relatively new flavor to recent active learning works, by considering active testing. In particular, they evaluate the learned model on the fly on adaptively acquired set, called quizzes. Model learns based on those quizzes and part of it is added, as a feedback from the quiz. At the end, it is evaluated on the final exam set (cumulative quizzes excluding feedback examples). Statistical convergence guarantees have been provided, and low variance (unbiased) estimators are considered. Proof of concept is provided on test dataset w/ GP model and common classification datasets are considered for the real world setting. The main paper provides the results on how far is true risk (estimated from large number of held out samples) from empirical risk.

**Strengths:**

The paper provides a relatively new flavor to recent active learning works, by considering active testing. In particular, they evaluate the learned model on the fly on adaptively acquired set, called quizzes. Model learns based on those quizzes and part of it is added, as a feedback from the quiz. At the end, it is evaluated on the final exam set (cumulative quizzes excluding feedback examples). Statistical convergence guarantees have been provided, and low variance (unbiased) estimators are considered. Theory and practical aspect seem interesting to the broad community. It is generally well written.

**Weaknesses:**

Much of the weakness in the work are some of my confusions and doubts. I *strongly* appreciate the authors to add an algorithmic block which shows how each and every step is done in the true implementation. Lastly, I haven't verified the correctness of the proof line by line, but the theorems seem sound.

On AL acquisition functions:
- I don't see any discussion at all about what AL algorithm is being used in all the experiments. Can authors please add that discussion.

On theory:

- How is the sampling from optimal $q_t$ done to get the quizzes at each testing round? The expression mentioned in equation (4) involves an integral done over squared error of loss function and true risk and the true underlying posterior distribution over labels. While for the true risk, it is approximated using a multi-source estimate, I am not sure where do we get the labels to even define the distribution.
- The multi-source estimate is a function of time step, that is ideally equation (5) should've $t$ in the LHS. Secondly, right after the first round, how are we getting $\hat{R}_{Q_t}$ to multi-source estimate?
- Wherever the expectation (integrals) are approximated using the empirical estimates in the quizzes or final test, or to get $C_t$, what is the scale of example? I feel that for small number of examples, $C_t$'s estimate can be pretty noisy, and therefore making $\tilde{R}$ estimate noisy.
- For the active feedback setting, is only 1 example chosen from the quiz set $\mathcal{Q}_t$? If not, how is the batch selection done? Equation (9) only seem to be providing information for 1 example. If it is more than that, then I'd appreciate writing this as an algorithm (greedy procedure), or at least some discussion.


On experiments:
- Can authors provide absolute accuracy numbers, and how does that change over the course of AL rounds?
- How is the scale of error in Fig 3g so different than the others?




**Questions:**

Please refer to the weakness section.

**Limitations:**

I feel that more dataset/arch should be considered to study the real impact of the early stopping method, and in particular it should be compared against (in a fair manner) to fix accuracy based criterion. While I appreciate the provided framework, it needs much more discussion as to how it would fit in the current paradigm, in particular when we have low budget active learning regime, or where we already start with a pretrained backbone (to be sample efficient). Lastly, I feel skeptic about the empirical estimates for the expectations, as usually the examples used for the testing would be small (as it counts in the labeling budget as well).

---

> ### Author Rebuttal · Authors · 2023-08-10
>
> **Q1: AL strategy specification.**
>
> Please see Q2 in the general response.
>
> **Q2: Sampling using $q_t$ and distribution estimation.**
>
> As we explained in Q1 in the general response, the test sampling is done in a sequential way. In each test sampling round, we compute $q({\bf x})$ over the entire unlabeled pool according to Eq. (4). We use a deterministic way of summing over all classes based on the posterior distribution predicted by the model. The multi-source estimate only utilizes labels of the training samples and previously selected test samples, and also uses the predictions for unlabeled samples. Later in the active feedback stage, if a test sample is selected to be added to the training set, we remove them from the testing set. Our newly provided ATL algorithm pseudo code also demonstrates the entire process (see our answer to Q3 in the general response).
>
> **Q3: Multi-source risk estimate.**
>
> Thank you for the suggestion, we will add $t$ to the terms in Eq.(5). In the first round, $\tilde{R}$ is not considered because there are no test samples yet. In each round since the first, all the terms in Eq.(5) are based on the current stage of the ATL process. Meaning that at time $t$, we will use the current training loss $R_{train}$, the current test risk $\tilde{R}$ (we will fix the notation in the paper, cannot add $t$ in openreview), and the currently computed $R_{\theta}$ (computed in the same way as in ref[19]).
>
> **Q4: Estimate of $C_t$ and scale.**
>
> Thank you for the question. As we find there is indeed a notation issue in the description of $C_t$, we would like to clarify that it should be $\frac{1}{n_t}\sum_{i=1}^{n_t}\sum_y\frac{p(x)}{q_t(x)}[\mathcal{L}(f_T(x),y)-R]^2p(y|x)$ ($R$ estimated by (R^{multi}{\theta}) at T). This definition is based on $\sigma_t(F_T)$ as previously mentioned, which scales with $n_t$. Although we do not consider the final $\tilde{R}$ noisy because of it, the number of test samples being small can indeed raise the concern of estimation quality, but this aligns with the fundamental challenge studied in this work. We have taken approaches to make the result as accurate as possible, e.g. the estimate of $(R^{multi}_{\theta})_T$ is done over the remaining unlabeled set. Since $v_t$ is always used after normalization, the key idea is to compare the confidence of quizzes of different $t$ and apply weights accordingly, serving the purpose of achieving the approximately variance-optimal result in the end.
>
> **Q5: Active feedback setting and algorithm box.**
>
> Please see Q3 in the general response about the algorithm. For test and feedback selections, we sequentially sample $n$ times to obtain each batch. The testing process does not involve re-training the model, thus the batch mode has no effect. For the feedback process, unlike AL where obtaining the label and re-training the model could have a large impact, we already have the labels for both training and testing samples at this moment. Re-training may have an impact, but is probably not worth the cost.
>
> **Q6: Test accuracy results.**
>
> Thank you for the suggestion. We here present the test accuracy results corresponding to the tables in the paper:
>
> | Dataset      | AL strategy | Feedback | Iter4  | Iter8  | Iter12  | Iter16  | Iter20  |
> | -------- | --------- | ----- | ----- | ----- | ----- | ----- | ----- |
> | MNIST | Entropy | NF | $83.6\%$ | $91.2\%$ | $93.2\%$ | $94.8\%$ | $95.9\%$ |
> | MNIST | Entropy | RF   | $83.3\%$ | $91.4\%$ | $94.2\%$ | $94.6\%$ | $96.1\%$ |
> | MNIST | Entropy | ATL | $84.3\%$ | $92.2\%$ | $94.3\%$ | $95.3\%$ | $96.1\%$ |
> | FashionMNIST | Entropy | NF   | $69.9\%$ | $77.0\%$ | $78.1\%$ | $81.5\%$ | $83.2\%$ |
> | FashionMNIST | Entropy | RF  | $73.0\%$ | $76.1\%$ | $78.6\%$ | $82.0\%$ | $83.3\%$ |
> | FashionMNIST | Entropy | ATL   | $72.9\%$ | $77.5\%$ | $80.8\%$ | $82.1\%$ | $84.3\%$ |
> | CIFAR10 | Entropy | NF  | $42.5\%$ | $46.8\%$ | $52.4\%$ | $57.0\%$ | $57.8\%$ |
> | CIFAR10 | Entropy | RF   | $42.1\%$ | $50.5\%$ | $49.1\%$ | $57.3\%$ | $58.2\%$ |
> | CIFAR10 | Entropy | ATL | $43.3\%$ | $50.4\%$ | $51.4\%$ | $56.8\%$ | $59.5\%$ |
> As we can see from the table, the accuracy results are mostly consistent with the true risk (lower loss coincides with higher accuracy) with few exceptions. We will further explain early-stopping related accuracy results in Q8: limitations.
>
> **Q7: Scale of error in Fig 3g.**
>
> Thank you for noticing the detail and raising the question. Figure 3 demonstrates the learning-testing-feedback process on a synthetic dataset. Because the dataset is simpler, the true risk of the model is very low, especially at a later stage as in quiz 18. The estimation error is also low in this case, particularly with the no feedback case (ATL-NF) as in Figure 3(g). However, in this case, there is no benefit to model training from the feedback.
>
> **Q8: Limitations**
>
> Thank you for the suggestion. We acknowledge that the current early stopping results have limitations as mentioned. However, the fixed accuracy stopping will result in a much higher variance in the stopping iteration.
>
> | Dataset      | Method | Iteration | Variance  | Test Accuracy  | Variance  |
> | -------- | --------- | ----- | ----- | ----- | ----- |
> | Fashion MNIST | Fixed | 15.8 | 1.36 | $81.33\%$ | $4.4e-5$ |
> | SVHN | Fixed |  16 | 1.6 | $85.20\%$ | $6.1e-5$ |
> | SVHN | Combined | 15.6 | 0.64 | $84.02\%$ | $3.2e-4$ |
>
> We agree that the expectations make the problem difficult. In this paper, we use the integration of quizzes, the improved intermediate estimate, and practical feedback solutions to address these challenges and conduct empirical studies. However, as the reviewer mentioned, there is still much room for improvement in the current framework. Again, we would like to re-state our contribution as proposing the first ATL framework, and presenting a practical solution that improves the learning performance while maintaining the quality of risk estimation given a limited total labeling budget.

---

> ### Author Response · Authors · 2023-08-18
>
> Thank you again for your review and suggestions. We hope that our responses have cleared the previous confusions. We appreciate the comments about the novelty of the problem and have incorporated your suggestions to further clarify the contribution as well as limitations. As we have also clarified the AL strategy and provided the missing results and necessary information in the rebuttal, we sincerely hope that you can find our answers satisfactory and consider updating the assessment. We are happy to provide any additional information if needed.

---

> > ### Comment · Reviewer_bQCr · 2023-08-21
> > **Thanks for the rebuttal.**
> >
> > I thank the authors for the rebuttal; I'd retain my rating.

---

> > > ### Author Response · Authors · 2023-08-21
> > >
> > > Thank you very much for reading our responses and keeping the positive rating. We will make sure to incorporate your suggestions in the revised paper.

---

### Official Review · Reviewer_LqNc · 2023-07-07

**Soundness:** 2 fair
**Presentation:** 2 fair
**Contribution:** 3 good
**Rating:** 6
**Confidence:** 3

**Summary:**

This paper proposes a framework for integrating active learning and active testing in an online fashion. The proposed algorithm incrementally sample training and testing examples for each batch and can leverage all of the testing examples so far for evaluation. Moreover, the sampling distribution for testing is designed based on an estimate of the variance of the risk. Experiments are conducted for both Gaussian processes and neural networks on multiple datasets.

**Strengths:**

The paper studies a very important problem and provides an effective solution. I find the method to be novel.

**Weaknesses:**

1. I am not sure if the claim that the test risk estimate is unbiased. It is indeed unbiased if one knows the true risk, but when using the estimate proposed in section 3.3 to substitute the true risk, is this still unbiased? Moreover, if the feedback set is used in training, the estimator doesn't seem to be unbiased either. I think the authors should play down the claim of the proposed risk estimate is unbiased since it is not when combined with all of the tricks.
2. In the experiments, the results of random sampling are not shown in the tables.
3. There are quite a few heuristics proposed in the paper from sections 3.3 to 3.5. Although the authors provided intuitive explanations, the design of the algorithm may have overfitted to the three datasets the authors tested on. For example, the weighting between different components of the intermediate estimate seems quite arbitrary (section 3.3). It is also not clear how C is chose in practice in section 3.5. I strongly encourage the authors to further test their design across a wider range of settings.
4. The paper assumes the active learning algorithm to sample based on a probability distribution. However, many practical AL algorithms do not. This can cause the test risk estimate to be really high for those algorithms. I strongly encourage the authors to address this problem.

**Questions:**

1. The active learning strategy used in this paper seems to be very vague. What's the algorithm being used?
2. There are quite a few components to the proposed algorithm. Can you give an algorithm box to make it clearer? Also, can you provide a time complexity analysis?
3. In [1] appendix , the authors used a water-filling algorithm to make sure the examples sampled up to time t follows roughly from distribution $q_t$, even though $q_1, ..., q_{t-1}$ may differ from $q_t$. I wonder how this technique compares against what's proposed in 3.4.
4. Please also see the weakness section. I think this work studies an important problem and proposes a cool solution. I am willing to raise my score if the authors sufficiently address (part of) my concerns.

[1] Katz-Samuels, J., Zhang, J., Jain, L., & Jamieson, K. (2021, July). Improved algorithms for agnostic pool-based active classification. In International Conference on Machine Learning (pp. 5334-5344). PMLR.

**Limitations:**

Sufficient.

---

> ### Author Rebuttal · Authors · 2023-08-10
>
> **Q1: The unbiased risk estimate.**
>
> Thank you very much for this insightful question! Please see the answer to Q4 and Q5 in the general response. Adding to Sections 3.2 to 3.4 in the paper, we discussed in detail the improved multi-source intermediate estimator (which impacts the variance-optimal solution but does not change the unbiased nature) in Q4, and the effect of feedback (for which we provide a practical solution and empirical studies) in Q5.
>
> **Q2: Random sampling results.**
>
> Thank you for raising this concern. We have already included the random feedback results in Table 3. In Appendix D.2.2 Table 6, we also provide the results using random test sampling results and detailed analysis. Please note that this result is quite different from the AT comparisons from previous works without the AL setting. First, the AL changes the proposal distributions and the straightforward ARE/AT/ASE integrates actually introduce biases. Second, the base model is not accurate enough, making the test sampling proposal less optimal. From the results, we can see that random test sampling is comparable to the baselines as it also asymptotically converges to the true risk, but ATL performs better.
>
> **Q3: Analysis from Sections 3.3 to 3.5 and the constant C.**
>
> We have further explained the problem setting in the general response. More specifically, we have established the quiz by using existing results in Section 3.2, improved upon existing intermediate estimate of the true risk in Section 3.3, theoretically constructed an unbiased low-variance integrate estimator in Section 3.4, and provided further analysis in Section 3.5. Most part of Section 3.4 and 3.5 are strict theoretical analyses using the properties of multi-variate Gaussian and other theoretical results. We have to introduce some heuristics in Section 3.3 and the later part of 3.5 to have practical solutions or discussions.
>
> The weighting between different components of the intermediate estimate is based on the size of each component at the current stage. Each component (training loss, predictive uncertainty, and test loss) is assigned a weight according to the number of samples we have. The idea is to improve upon only using $R_{\theta}$ since $R_{\theta}$ usually underestimates the true pool risk (in a poor-calibrated case, overestimating is also possible). We agree that the current $R^{multi}_{\theta}$ is still heuristic, but in future works we might further improve the estimate using some theoretical insights (which usually require more assumptions to be made about the model and AL). In Section 3.5, the constant $C$ is not a specific parameter, and is not utilized in the practical solution. It is used for the purpose of theoretical analysis, and the other part that matters is that in general settings it can be inferred to be a constant of $\mathcal{O}(1)$, meaning that a balance can indeed be achieved between the two components in the joint learning-testing objective.
>
> **Q4: Practical AL algorithm and setting clarification.**
>
> We would like to clarify that we work in a setting where the unlabeled pool is significantly larger than the labeled training set. Such a setting is commonly adopted that covers many practical scenarios where the labeling budget is limited or the labeling cost is prohibitive.
> The proposed risk estimate does not depend on (or make assumptions about) the AL sampling distribution. We agree that practical AL strategies might not have clearly defined sampling distributions, but given the scale of our problem setting, we assume the test risk estimation to be mostly orthogonal to the AL. As we explained about the unbiased risk estimation (see the answer to Q4 in the general response), the estimation error is expected to be low when we use the approximately variance-optimal test sampling approach.
>
> **Q5: AL strategy specification.**
>
> Please see Q2 in the general response.
>
> **Q6: Algorithm box and time complexity.**
>
> Please see Q3 in the general response. As for the time complexity, we should study each component separately as the number of samples varies. The AL and training components are standard. In test selection, we first compute $R_{\theta}$ ($\mathcal{O}(N_U)$), then evaluates $q(\bf x)$ for the unlabeled pool ($\mathcal{O}(N_U)$). The integrated risk estimation takes $\mathcal{O}(N_UTN_T)$. The feedback might require an additional step of computing the distance ($\mathcal{O}(N_TN_L)$). The sampling complexity should be comparable to AT/ASE methods, which require additional complexity $\mathcal{O}(N_LTN_T)$ for training surrogate models.
>
>
> **Q7: Comparison with reference [1].**
>
> Thank you for suggesting the reference! The method mentioned is a very interesting approach to achieving optimal sampling in multiple rounds. However, since the scope of the method is only on active learning, full replacement is allowed and the entire sampling serves the single purpose of ensuring the learning performance. In our setting, since our test sampling follows AL sampling and model training, the change of target is much more significant between rounds. As we show in Section 3.3, the sampling in each quiz is theoretically variance-optimal for the current model. However, as the model changes, the goal also changes, and we expect the sampled data instances needed to compensate for the change will be greater than the pure-AL case, meaning more labels might be required to achieve a globally optimal sampling result than the locally optimal one. Unfortunately, due to the vast difference in settings (AL vs ATL, binary vs general, among others), it is not likely we can provide a reasonable comparison between the methods. Our integration in Section 3.5 has a similar effect to adjusting to the global optimal (but using the fixed budget for each round) by adding another layer of weights to the local optimal according to a confidence-based metric.

---

> ### Author Response · Authors · 2023-08-18
>
> Thank you again for your review and suggestions. We hope that our responses have clarified the confusions raised in the review. We have incorporated your suggestions and further clarified the contribution including the unbiased risk estimator for integrated quizzes over AL and the high-level analysis plus practical solution for active feedback. As we have also clarified the AL strategy and provided the missing results and necessary information in the rebuttal, we sincerely hope that you can find our answers satisfactory and consider updating the assessment. We are happy to provide any additional information if needed.

---

> > ### Comment · Reviewer_LqNc · 2023-08-20
> >
> > I would like to thank the authors for their detailed rebuttal. Most of my concerns are addressed and I have raised my score. I would like to suggest the authors to distinguish what is heuristic versus principled by restructuring section 3. Currently, it is not clear which of the estimates are unbiased. Perhaps stating out the biased and heuristics could really help the reader to understand. I believe we should not shy away from admitting things are heuristic in our papers.

---

> > > ### Author Response · Authors · 2023-08-20
> > >
> > > Thank you very much for the response. We are happy to find that we have addressed most of your concerns. We will carefully follow your suggestions to update the paper. We will further clarify in Section 3 to distinguish what is the unbiased integrated estimator versus the heuristics used in the practical solution.

---

### Author Rebuttal · Authors · 2023-08-10

In this general response, we address a set of important questions that commonly occur in multiple reviewers' comments, avoiding repeating the same response in each individual rebuttal.



**Q1: Clarification of the problem setting. (To all reviewers)**

In this paper, we aim to construct an integrated framework of active-testing-learning. This problem is a novel but practical one. Unlike many active learning (AL) or few existing active testing (AT) works that focus on efficient sampling designed solely for training or testing, respectively, the proposed work is the first effort that simultaneously considers both active testing and learning using an integrated framework. It tackles a much more challenging but highly practical setting, where the total labeling budget for the entire learning-testing (validation) process is limited. In the scope of the proposed work, we focus on studying risk estimation and the active feedback process that can work seamlessly with a wide range of different AL algorithms (we will elaborate on the AL-agnostic related results below). Following this rationale, our evaluation mainly covers the following perspectives: (1) for the first objective, we compare the estimated risk with a hold-out test risk (representing the true risk); (2) for the second objective, we evaluate how well we can maintain an accurate risk estimation while improving the model performance using active feedback.

**Q2: AL strategy. (To reviewers LqNc, bQCr, and 5t4X)**

The ATL framework is designed to be AL-agnostic so that it can be easily integrated with a wide range of commonly used AL algorithms. To demonstrate its general applicability, the main results are obtained using uncertainty (i.e., entropy) based sampling as the AL strategy given its great popularity in many AL models. We will more clearly describe and explain the chosen AL strategy in the revised paper. Additionally, we have gathered risk estimation results using two commonly used AL strategies: *margin sampling* (Wang et al. A new active labeling method for deep learning), *BALD* (Houlsby et al. Bayesian Active Learning for Classification and Preference Learning).

The results (in PDF) show that the integrated risk estimation performs similarly regardless of the AL strategy. The relative scales of different risk estimations are also similar to the entropy case in the main paper. As for feedback results, we do see some different effects. Depending on the AL strategy, random feedback performs unstably. However, our proposed feedback still mostly outperforms the fair comparison baseline.





**Q3: Algorithm block. (To reviewers LqNc and bQCr)**

Following the reviewers' suggestion, we have provided an algorithm block (in the attached PDF) that includes and organizes all the key steps in a systematic way.

**Q4: The unbiased risk estimator and the multi-source estimate. (To reviewers LqNc and bQCr)**

The unbiased risk estimation achieved by each quiz is guaranteed asymptotically by Lemma 1 in (Active risk estimation [19]), which we referenced in Eqs. (2)(3). Then, we extended the unbiased quiz-wise risk estimator to the integrated version (Section 3.4) through the multivariate Gaussian analysis before Theorem 2 in our paper. Since the asymptotic relationship is not great when $N$ is small, the need for reducing the variance of the estimation becomes even more important in the AL setting. The key idea of Theorem 1 is to analyze the best variance of the integrated estimator given quizzes. To achieve this, the test sampling proposal in each quiz is kept as in Eq. (4).

Guided by our theoretical analysis, the multi-source estimate $R_\theta^\text{multi}$ of the true risk is introduced to construct the test sampling proposal distribution $q$ in practice. It is an improvement upon the $R_\theta$ proposed by Christoph et al. (Active risk estimation [19]). Ideally, if $R_\theta$ is close to the true risk $R$, the test sampling proposal will have the lowest variance while unbiasedly converging to the true risk. Similarly, in Active Testing ([13]), if the true risk is known, the optimal test sampling proposal can be directly constructed using the individual losses, resulting in an immediately converging estimator. However, in reality, the labels are unknown for the samples to be selected. Thus, an approximation has to be made and will affect the quality of risk estimation. We would like to clarify (similar to [13, 19]) that since the estimators are weighted (by importance weighting or AT-unbiased weighting) sums, they always asymptotically converge to the true risk (in the same way as random sampling will also converge to the true risk, just slowly). Thus, the quality of the estimation really depends on the variance of the estimator. Again, the true variance can not be known beforehand because we do not have access to the labels, thus the practical solutions are proposed to achieve good testing sampling. In Proposition 1, we show that existing $R_\theta$ is not optimal. By combining the current loss information that we do have access to, including ($R_{train},R_{\theta},\hat{R}_{Q_t}$), it allows us to bring the intermediate estimate closer to the true risk.



**Q5: The effect of feedback on the unbiased risk estimator. (To reviewers LqNc and 2XK2)**


We want to clarify that our paper explicitly claims unbiasedness only for the integrated estimator $\tilde{R}$. The active feedback process may indeed affect the unbiased risk estimator in a negative way. We show that the negative impact could be controlled in a combined objective of learning-testing. In a general analysis, we show that active feedback could indeed achieve an optimal solution for the joint optimization problem. Then we provide a practical solution and empirically verify the theoretical by showing that active feedback can indeed improve model learning without sacrificing the risk estimation too much. Please see the answer to Q5 from reviewer 2XK2 for more details.

---

### Decision · Program_Chairs · 2023-09-21

**Decision:**

Accept (poster)

**Comment:**

This paper introduces the "ATL framework" that integrates active learning with model testing, The goal is to perform unbiased risk estimation. The reviewers noted the novelty and importance of the approach/problem. There is expectation that this will be of interest to the NeurIPS community.